# Drosha drives the formation of DNA:RNA hybrids around DNA break sites to facilitate DNA repair

Wei-Ting Lu [1], Ben R. Hawley [1], George L. Skalka[1], Robert A. Baldock[2,3], Ewan M. Smith[1], Aldo S. Bader[1], Michal Malewicz[1], Felicity Z. Watts[2], Ania Wilczynska [1] & Martin Bushell[1]

The error-free and efficient repair of DNA double-stranded breaks (DSBs) is extremely important for cell survival. RNA has been implicated in the resolution of DNA damage but the mechanism remains poorly understood. Here, we show that miRNA biogenesis enzymes, Drosha and Dicer, control the recruitment of repair factors from multiple pathways to sites of damage. Depletion of Drosha significantly reduces DNA repair by both homologous recombination (HR) and non-homologous end joining (NHEJ). Drosha is required within minutes of break induction, suggesting a central and early role for RNA processing in DNA repair. Sequencing of DNA:RNA hybrids reveals RNA invasion around DNA break sites in a Drosha-dependent manner. Removal of the RNA component of these structures results in impaired repair. These results show how RNA can be a direct and critical mediator of DNA damage repair in human cells.

[1] MRC Toxicology Unit, Lancaster Road, Leicester LE1 9HN, UK. [2] Genome Damage and Stability Centre, School of Life Sciences, University of Sussex, Brighton BN1 9RQ, UK. [3] Present address: University of Pittsburgh Cancer Institute, University of Pittsburgh, Pittsburgh PA 15232 PA, USA. Wei-Ting Lu and Ben R. Hawley contributed equally to this work. Correspondence and requests for materials should be addressed to M.B. (email: mb446@leicester.ac.uk)

High fidelity DNA replication and repair is of paramount importance for the maintenance of genome integrity[1]. Intrinsic processes and extrinsic factors can result in several different forms of DNA damage, the most deleterious of which are double strand breaks (DSBs). The DNA damage response (DDR) consists of damage recognition, followed by signaling and chromatin remodeling cascades. These allow the recruitment of various repair proteins to the site of damage, which ultimately results in resolution of the DSB[1]. Several key events define this process. First, a feedback loop involving the kinase ATM (ataxia-telangiectasia mutated), MDC1 (Mediator of DNA Damage Checkpoint 1), and the MRN complex (Mre11-Rad50-NBS1 complex) leads to the propagation of phosphorylated histone H2A.X ($\gamma$H2A.X). Next, chromatin remodeling and modification factors are recruited in a $\gamma$H2A.X-dependent manner to open up the chromatin and deposit various modifications that allow recruitment of a series of effector proteins. The repair of the DSB then takes place via one of two main pathways: error-free homologous recombination (HR) or error-prone non-homologous end-joining (NHEJ). A critical factor for pathway selection is the accumulation of 53BP1 (tumor suppressor p53 binding protein 1) at DSB sites, which limits the extent of resection from the break to promote NHEJ[1,2]. In S/G2 phase, Breast Cancer 1 (BRCA1), which is considered antagonistic to 53BP1[3], is recruited and promotes DNA end-resection and HR resolution.

While most investigations have focused on the proteins involved in DDR, the involvement of RNA molecules has only recently been investigated. For example, a number of high-throughput screens have highlighted the critical involvement of RNA processing enzymes in DNA repair[4–6]. Moreover, a recent study demonstrated that the choice of repair pathway is dictated by the transcriptional status of the damaged locus[7], further implicating RNA processing in DNA repair. Of particular interest is the involvement of the microRNA (miRNA) biogenesis apparatus, specifically the RNase III enzymes Drosha and Dicer, in genomic stability[8,9] and DNA repair[10]. The canonical activity of these proteins involves the maturation of miRNAs, where sequential cleavage events by Drosha in the nucleus and Dicer in the cytoplasm generate short double stranded RNA (dsRNA) molecules from their primary transcripts[11]. These are then loaded in the Argonaute-containing RNA-induced silencing complex (RISC) which identifies target mRNAs by imperfect base pairing and silences protein production through the interaction of RISC with the TNRC6 family of proteins[12,13] (Supplementary Fig. 1A). Whilst the mechanism through which Drosha and Dicer affect DNA repair remains unknown[10], small RNAs arising from the sequences around DSBs have been observed and appear to have a function in DDR[14,15]. Different mechanisms for the role of these small RNAs have been proposed: acting as bona fide endo-siRNAs to silence aberrant RNA produced from damaged DNA[16–18], or by facilitating sequence-specific recruitment of repair factors to sites of damage[14,15,19]. The biogenesis and mechanism of action of these small RNAs, as well as whether they are a general feature of all DSB repair remains unknown. Interestingly, both Drosha and Dicer have additional functions that are independent of their activities in the generation of small RNA[8,20,21]. They have been shown to play a critical role in the non-canonical termination of mRNA transcription[8,20]. Furthermore, components of the RNAi machinery, particularly Dicer, have been critically linked to the processing of DNA:RNA hybrid structures at RNA polymerase II (Pol II) termination sites[8]. Moreover, nuclear Dicer is reported to be required to limit the generation of deleterious double-stranded RNAs in the nucleus[21].

Here, we systematically investigate the role of Dicer and Drosha in DDR, particularly in relation to RNA in the proximity of DSBs. We take advantage of an inducible restriction endonuclease system that introduces double strand breaks at a number of different genomic loci and show that small RNA generation is not a wide-spread phenomenon in DDR. Using a high-throughput approach we uncover the formation of DNA:RNA hybrids at break sites and demonstrate that their formation is dependent upon Drosha.

## Results

**Drosha and Dicer are required for 53BP1 foci formation**. To determine the relative contribution of the miRNA pathway components (Supplementary Fig. 1A) in DDR, we investigated the formation of 53BP1 foci after a 6 h recovery from ionizing radiation (IR). Depletion of Drosha, Dicer and TNRC6A-B did not prevent the initial sensing of DNA breaks, as indicated by the phosphorylation of histone variant H2A.X ($\gamma$H2A.X) and the checkpoint protein Chk2 (Fig. 1a–c, Supplementary Fig. 1B)[1]. The knockdown of Drosha and Dicer significantly reduced the recruitment of 53BP1 to damage sites, a critical step required for the subsequent activation of the DNA repair pathway (Fig. 1a–c)[1]. Since TNRC6A-B knockdown does not affect 53BP1 foci formation, it is likely that the canonical miRNA repression pathway is not involved in this process. To additionally confirm that miRNAs are unaffected, we examined the level of mature miR-21 upon Dicer, Drosha and TNRC6A-B knockdown (Supplementary Fig. 1C). As expected, given the longevity of some small RNAs[22], the depletion of the small RNA biogenesis enzymes for 48 h did not result in changes in mature miRNA levels explaining why Drosha/Dicer depletion does not affect miRNA-mediated repression (Supplementary Fig. 1E). Furthermore, whilst TNRC6A-B knockdown does not affect 53BP1 foci formation after IR irradiation (Fig. 1a–c), activity of the canonical miRNA pathway was significantly diminished only after TNRC6 knockdown (Supplementary Fig. 1D, E). This confirmed that in the experimental timeframe, the observed effects are likely not linked to the canonical function of the miRNA repression pathway. Taken together our data show that miRNA function is not required for alterations in 53BP1 recruitment and that the involvement of miRNA biogenesis enzymes in DDR stems from an additional activity.

**Drosha is required early in the DNA damage response**. As the observed impact on 53BP1 recruitment was consistently greater following Drosha depletion compared to Dicer knockdown (Fig. 1a, b), we focused primarily on the role of this protein in DNA repair. Previous reports had suggested different stages in the DSB repair pathway were affected by Drosha[14,19,23], thus we employed a systematic approach to determine what stage of the repair pathway Drosha acts upon, and whether both error-free HR and error-prone NHEJ were affected by its depletion. To address this, we allowed A549 cells to recover for a shorter time (2 h) from IR exposure. The knockdown of Drosha did not affect early DNA damage signaling components upstream of 53BP1, including H2A.X phosphorylation, ATM or MDC1 recruitment (Fig. 2a, b, Supplementary Figs. 2 and 3A)[1]. Thus, immediate signaling propagation around the break site is unaffected by the absence of Drosha. 53BP1 retention at DSB sites, which is dependent on Drosha (Fig. 1, Supplementary Fig. 3B), is required for proficient NHEJ[1,2,24]. In HR, BRCA1 competes with 53BP1 and its partner Rif1 at DSB sites[24,25] and in the absence of 53BP1, HR factor recruitment is increased[26,27]. During HR, the presence of BRCA1 promotes resection of DNA at DSBs[3,25]. The resected single stranded ends are then bound by Rad51 recombinase to mediate strand exchange[1]. To determine the role of Drosha in repair pathways downstream of 53BP1, we examined BRCA1 and

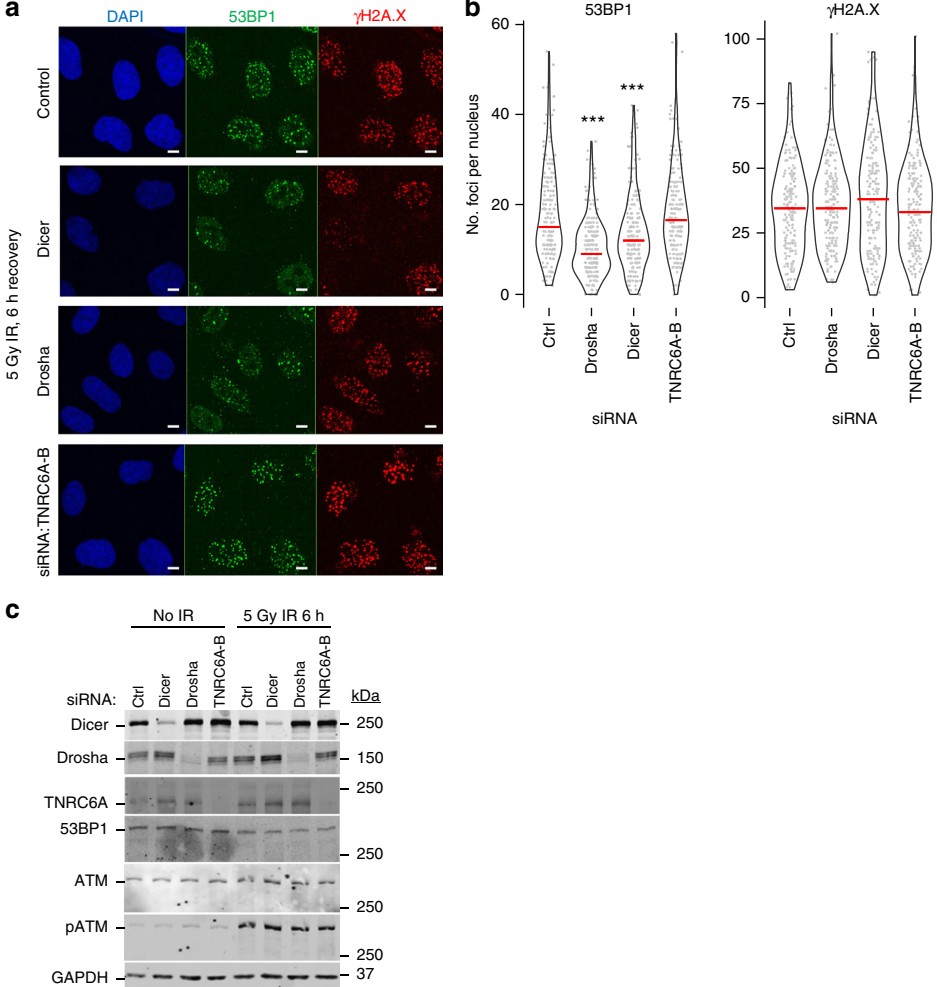

**Fig. 1** Dicer and Drosha are required for formation of 53BP1 foci after ionizing radiation (IR). **a** Representative immunofluorescence images visualizing IR-induced 53BP1 foci in A549 cells. 53BP1 in green channel, γH2A.X in red, DAPI-stained nuclei in blue. Scale bars, 10 μm. **b** Quantification of **a**. The number of foci per nucleus was counted using the FindFoci ImageJ plugin and plotted as individual data points (gray) and a violin plot. Data from 3 biological replicates, counting at least 60 cells per replicate. Red line represents the median in each condition. Statistical testing was performed using Dunn's test with Bonferroni correction for multiple comparisons, ***$p \leq 0.001$. **c** Representative western blots for **a**, **b** confirming knockdowns and induction of DNA damage

Rad51 foci formation upon Drosha knockdown. We observed a reduction in both BRCA1 and Rad51 foci, which suggested the HR pathway may also be impaired (Fig. 2a, b, Supplementary Figs. 4 and 5A). It is well known that the selection of HR or NHEJ for repair of DSBs is largely dependent on the phase of the cell cycle[1,2], however, we observed no alteration in the cell cycle following the depletion of Drosha (Supplementary Fig. 5B). This indicates that the lack of recruitment of HR and NHEJ markers to break sites upon Drosha knockdown is not cell cycle related. In summary, these data suggest that the activity exerted by Drosha lies downstream of ATM phosphorylation, cell cycle checkpoint activation (Supplementary Fig. 1B), and MDC1 localization, at the chromatin remodeling stage[2] of the repair pathway.

DNA repair normally commences rapidly following break site recognition, yet previous reports on the activity of Drosha in DDR had primarily focused on events hours after damage[10,14,15]. Having determined that the role of Drosha in DDR lies directly downstream of the very early signaling events, we investigated the dynamics of DDR following Drosha depletion. Allowing a 30 min recovery time after IR treatment revealed a significant decrease in 53BP1 recruitment after Drosha depletion (Fig. 3a, b). To evaluate the dynamics in more detail, cells stably expressing

GFP-tagged 53BP1 were subjected to laser microirradiation[28], which induced localized DNA damage, and the recruitment of 53BP1 to these sites was measured in real time (Fig. 3c, d). The depletion of Drosha significantly reduced the speed of redistribution of 53BP1 to break sites as early as a few minutes following damage (Fig. 3c, d). Thus, Drosha is required in DDR upon or directly before recruitment of 53BP1 (Supplementary Fig. 5A). Consolidation of our data from different time points confirmed that impairment of 53BP1 recruitment is an early event and continues for long periods of time following exposure to DNA damage (Supplementary Fig. 5C). Accordingly, Drosha and Dicer depletion increases the entry into late apoptosis following radiomimetic bleomycin-induced DNA damage (Supplementary Fig. 5D). Our observations clearly show that Drosha and Dicer have a direct role in the DNA repair process, distinct from that of the canonical miRNA pathway. In addition, we have been able to observe in real time its effects on the recruitment of the DNA damage repair factors.

**Drosha is required for effective HR and NHEJ.** While formation of foci at DSBs is indicative of recruitment of specific repair factors, they provide little information about the actual outcome

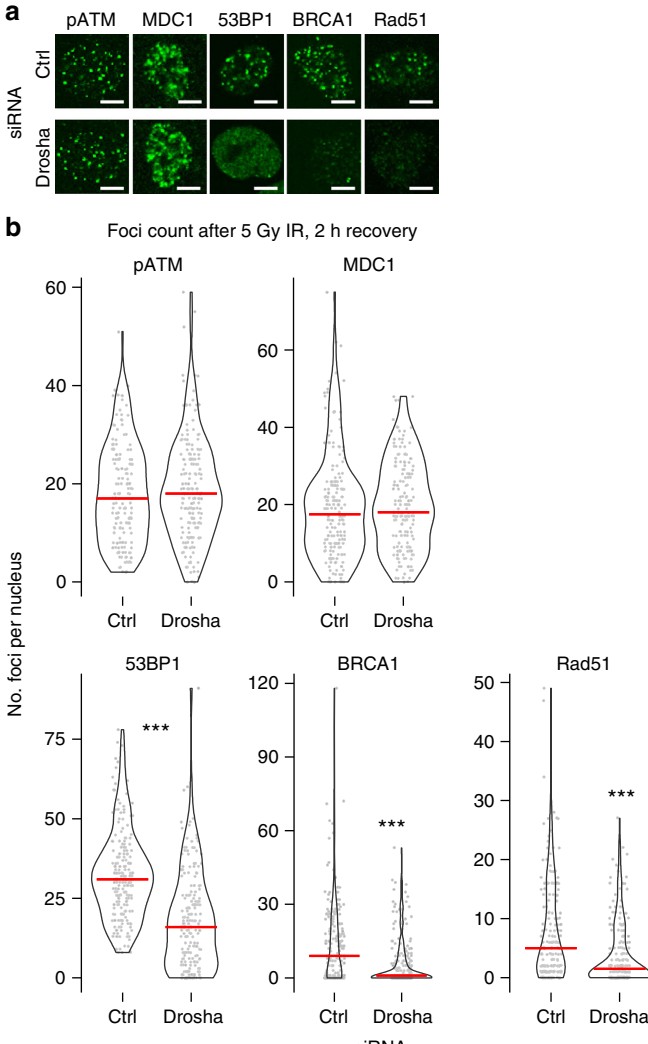

**Fig. 2** Drosha is involved in DDR downstream of MDC1. **a** Representative immunofluorescence image visualizing IR-induced DDR foci in A549 cells, 2 h post 5 Gy IR. Scale bars, 10 μm. **b** Quantification of **a**. Number of indicated foci per nucleus was counted as in Fig.1b across 3 biological replicates counting at least 60 cells per replicate. Red line denotes the median in each condition. Statistical significance determined using Mann-Whitney non-parametric test, ***$p \leq 0.001$

of the DNA repair. To evaluate the impact of Drosha depletion on error-free HR repair, we turned to a GFP-based HR repair reporter (Fig. 4a)[29]. In this stable U2OS cell line system, the expression of the endonuclease I-SceI results in the cleavage (and thus a double strand break) of a frame-shifted and therefore inactive GFP. Upon HR repair of this break, the frame-shift is corrected, using a corresponding in-frame GFP fragment present downstream in the reporter construct as a template. This allows expression of functional GFP which can be used as a measure of HR repair efficiency. The depletion of Drosha and Dicer severely compromised HR efficiency, similar to the effect seen upon depletion of BRCA1 (Fig. 4b). Meanwhile, the knockdown of TNRC6A-C had no effect on HR in this reporter system. Similarly, a GFP reporter system, in which NHEJ-mediated repair of I-SceI cleavage leads to measurable expression of the reporter protein[29] (Fig. 4c), showed that Drosha and Dicer, but not TNRC6A-C, are also involved in the NHEJ pathway (Fig. 4d). This is consistent with our findings that the miRNA biogenesis enzymes function upstream of the divergence of the two

pathways. Again, these results demonstrate that the RISC complex-mediated miRNA pathway is not required for HR or NHEJ repair and emphasizes a non-canonical role of miRNA biogenesis enzymes in these processes.

As mentioned above, the phase of the cell cycle was believed to be the major determinant of repair pathway choice[1,2], however, recent findings have shown that transcriptional status is also important in determining the repair pathway[7]. It was proposed that highly transcriptionally active sites are repaired mainly via HR, while more transcriptionally dampened areas of the genome are primarily processed by NHEJ. This may reflect a requirement for particularly high repair fidelity around transcriptionally active regions of the genome. The authors of Aymard et al.[7] used an elegant system to induce DNA DSBs at multiple endogenous genomic loci by introducing the DNA endonuclease AsiSI enzyme that translocates into the nucleus of U2OS cells inducibly upon 4-hydroxytamoxifen (4-OHT) treatment. They were able to map the endogenous cut sites and determine which individual sites were predominately utilizing either HR or NHEJ thanks to preferential association of Rad51 with HR sites[7]. It should be noted that only a subset of possible AsiSI recognition sites in the genome were shown to be actually cut, most likely due to their accessibility and methylation status[7].

The AsiSI system allows the opportunity to examine a number of well-characterized sites with known transcriptional status in the genomic environment. Using this system, we confirmed that depletion of Drosha and Dicer, but not TNRC6A-B, has the same impact on recruitment of repair factors to DNA damage foci as was observed following IR in A549 cells, and again this is not dependent on the miRNA pathway or cell cycle (Supplementary Figs. 6A, B and 8A, B, C). We confirmed that these effects were unchanged using different siRNAs against Drosha and Dicer (Supplementary Fig. 7A–F). Rescue experiments using an siRNA-resistant over-expression plasmid showed that the effects on 53BP1 recruitment were specific to Drosha knockdown (Supplementary Fig. 7G, H, I). Additionally, the reduction in Rad51 foci in this system reaffirmed the role for Drosha in both major repair pathways (Supplementary Fig. 6C, D). The recruitment of the E3 ubiquitin ligase RNF168 and local ubiquitination at DNA damage foci was also reduced upon depletion of Drosha (Supplementary Fig. 6E–H), thus strengthening the conclusion that Drosha acts at the chromatin remodeling phase prior to 53BP1 recruitment. We also confirmed that depletion of Drosha does not alter the cleavage efficiency within the inducible system (Supplementary Fig. 9).

We next sought to assess the impact of Drosha on DNA resection following induction of DSBs at specific genomic loci using the AsiSI system. Commitment to the HR pathway results in resection of DNA around the break site to allow the invasion of the homologous sister chromatid for priming of templated repair. Following extraction of genomic DNA, in vitro digestion at specific sites surrounding DSBs using restriction enzymes will cleave only dsDNA, but not single stranded DNA (ssDNA). The progression of resection can therefore be monitored by a qPCR-based approach by designing PCR primers spanning digested sites that allow amplification only where uncut resected ssDNA is present (Fig. 5a)[30,31]. As expected, end-resection at transcriptionally active HR-prone sites, as defined in ref.[7], was observed in a distance-dependent manner around the break site (Fig. 5b)[30]. The depletion of Drosha significantly reduced the percentage of resected DNA and the extent of resection from the break site (Fig. 5b). This could be rescued by the over-expression of an siRNA-resistant Drosha (Supplementary Fig. 10A), corroborating results of the rescue experiment which showed a restoration of 53BP1 foci (Supplementary Fig. 7G, H, I). Meanwhile, no increase in ssDNA around AsiSI-induced NHEJ-repaired break sites was

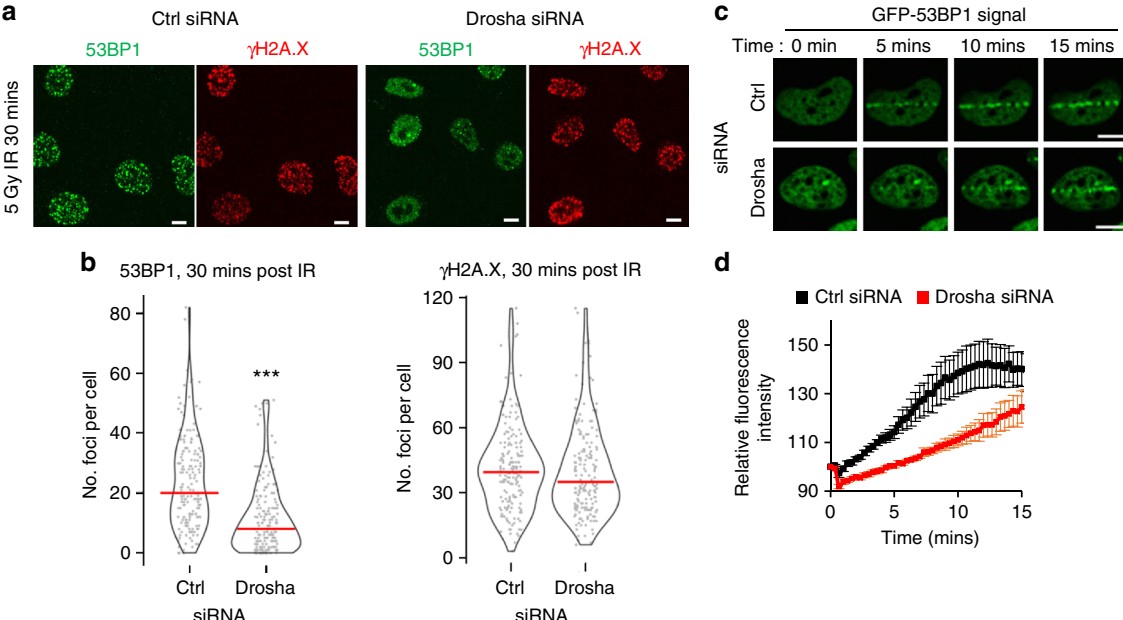

**Fig. 3** Drosha is required for early 53BP1 localization after DNA damage. **a** Representative immunofluorescence image of IR-induced 53BP1 foci in A549 cells, 30 mins post 5 Gy IR. Scale bars, 10 μm. **b** Drosha knockdown significantly impairs 53BP1 foci formation as early as 30 minutes after 5 Gy IR. Violin plots and data points show quantification, as in Fig. 1B, of IR-induced 53BP1 and γH2A.X foci shown in **a**. 200 cells were counted across 3 biological replicates, ***$p \leq 0.001$, Mann-Whitney non-parametric test. **c** U2OS cells expressing GFP-53BP1 were subjected to laser microirradiation and GFP redistribution was monitored in real time. Images of selected time points post microirradiation are shown. **d** Quantification of time-course as in **c**, error bars = SEM, $n \geq 30$ cells per condition over 4 replicates

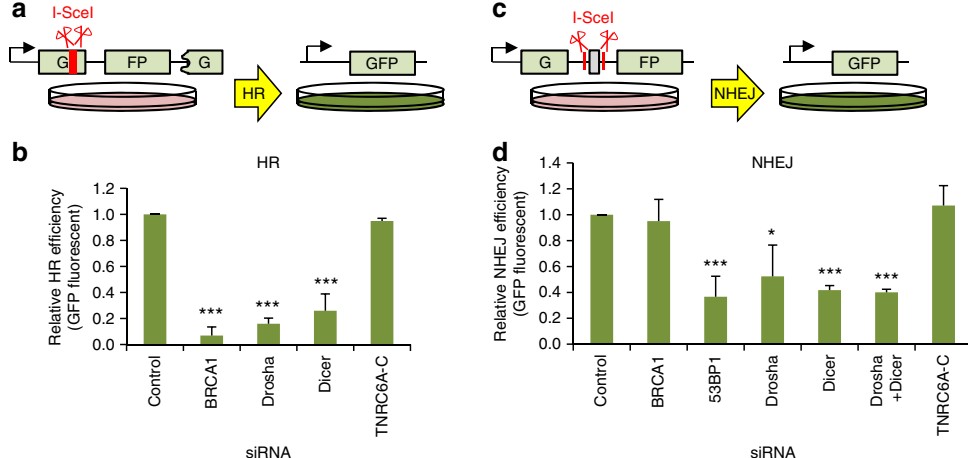

**Fig. 4** Drosha is required for effective homologous recombination and non-homologous end joining. **a** Cartoon depicting the quantitative HR reporter assay. Incompatible DNA ends are generated after digestion of the chromosomally integrated reporter with I-SceI endonuclease. Gene conversion after HR repair reconstitutes an active GFP gene, measured by FACS. **b** Quantitation of HR repair efficiency in U2OS cells using the reporter system described in **a**. In all cases, error bars = SD, Wilcoxon signed rank non-parametric test, ***$p \leq 0.001$, $N = 3$. **c** NHEJ reporter schematic. An adenovirus exon results in a non-functional GFP product; following cleavage by I-SceI, this is excised and NHEJ repair ligates introns 1 and 2 together. Splicing of the NHEJ-repaired product results in an active GFP which can be measured by FACS. **d** Quantitation of NHEJ repair efficiency in U2OS cells using the reporter system described in **c**. Mean of 3 biological replicates, error bars = SD, Wilcoxon signed rank non-parametric test, *$p \leq 0.05$, ***$p \leq 0.001$

observed in the presence or absence of Drosha (Supplementary Fig. 8D, E). We also used this assay to confirm that knockdown of Dicer had a similar effect on resection as Drosha depletion (Supplementary Fig. 10B, C). The two proteins seem not to have redundant functions in this pathway, as a double knockdown of Drosha and Dicer together has the same effect on resection as depletion of only one of the proteins (Supplementary Fig. 10B, C).

Since Drosha depletion had a significant effect on resection at HR sites, we asked if it is recruited to them, as this would allow it to contribute directly to the repair process. Drosha chromatin immunoprecipitation (ChIP) was followed by qPCR at an HR-specific AsiSI cut site and a genomic region lacking AsiSI recognition motifs, the miR-122 locus, previously shown to recruit Drosha[20]. This showed a robust increase of Drosha

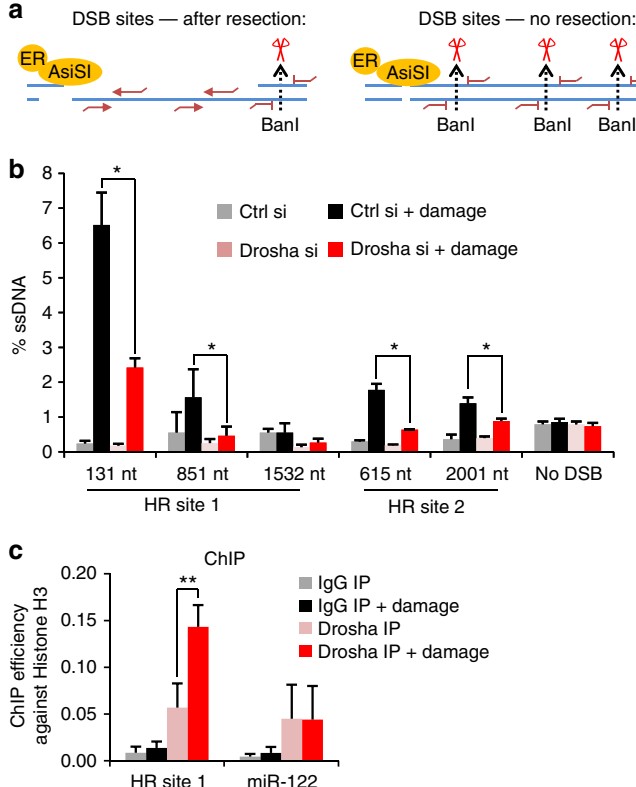

**Fig. 5** Drosha is required for DNA resection after DNA damage and is recruited to DSBs. **a** Cartoon describing quantitative DNA resection assay. Treatment of U2OS cells with 4-OHT induces damage by HA-ER-AsiSI. Genomic DNA is harvested and subjected to restriction digest (dotted lines). qPCR primers, red lines, are designed at either side of the restriction sites. Only loci that had been resected prior to digest can be amplified. **b** qPCR resection assay as in **a** at two HR sites, and the control locus (no DSB) which does not span an AsiSI restriction site, as described in ref. [7] following 4 h of damage induction. Error bars = SD, Student's 2-sample *T*-test, *$p \leq 0.05$. **c** qPCR of Drosha and control IgG ChIP at an HR locus and a canonical Drosha binding site at the miR-122 genomic locus 1 h after damage induction. The ChIP efficiency was calculated against a histone H3 ChIP performed in parallel, error bars = SEM, Student's 2-sample *T*-test, **$p \leq 0.01$, $N = 3$

interaction with the DSB following damage induction, while no change was observed at the control miR-122 locus (Fig. 5c). Together, these data suggest that Drosha plays a direct role in the resolution of DNA damage at sites undergoing HR repair.

**No significant increase of small RNAs around DSB sites**. Previous reports have shown production of small RNAs from sequences around DSBs, with the conclusion that these play a role in the DNA damage response[14–16]. As these sequencing experiments used highly expressed reporter systems[15,16] or break sites flanked by repetitive sequences[14] (or more recently the [TTAGGG]$_n$ telomere repeat sequence[32]), we set out to determine whether DSB-specific small RNAs are a more general feature of DNA repair using the AsiSI system. We examined 99 loci in the endogenous genome which have previously been shown to be cut robustly and reproducibly by the restriction endonuclease[7]. As Drosha is required very early after damage (Fig. 3d), we would expect that any functional small RNAs produced would be present at these earliest time points. Previously, however, the earliest time point at which the small RNAs involved in DDR were examined was 12 h following damage induction[14,15]. To conduct

a comprehensive investigation, RNA was harvested for high-throughput small RNA sequencing at time points ranging from 1 to 24 h following induction of DNA damage (Supplementary Fig. 11A, B). We compared the quality of our data to previously published high throughput experiments reporting the existence of DSB-derived small RNA[14]. As the previously published experiments had been performed in mouse cells and ours in human cells, we were only able to directly compare conserved mature miRNAs. Our experiments show at least equal coverage and depth (Supplementary Fig. 11D, E) as that of the previous study, thus we were confident that small RNAs arising from DSBs should be robustly detected. We observed no evidence of specific small RNA products being generated around any of the 99 cut sites at any time point (Fig. 6, Supplementary Fig. 11C). While we do observe a minor degree of variability at different time points, this is not statistically significant at any site or time point studied. In addition, examination of uncut loci with the same transcriptional activity as the AsiSI sites shows a similar degree of variation (Fig. 6 right hand panels). We also examined the size distribution of RNAs sequenced in our experiment (Supplementary Fig. 11F), which confirmed that the vast majority of small RNAs corresponded to known annotated small RNAs. We additionally analysed RNAs both smaller (19–20 nt) and larger (24–26 nt) in size than the canonical 21–23 nt, but this did not reveal any DSB-linked sequences over control conditions (Supplementary Fig. 12). Finally, we excluded the possibility that the small RNAs could be generated with a 5′-triphosphate (a feature described in plants and lower eukaryotes[33,34]) (Supplementary Fig. 13). These results suggest that previously reported DNA damage-induced small RNA may be specific to DNA damage in repetitive sequences[14,32] or a by-product of highly over-expressed loci[15,16], potentially due to the non-specific degradation of existing RNAs. Indeed, a recent publication questioned the production of damage-induced small RNAs in plants suggesting their initial discovery may have resulted from analysis of particular reporter systems[35].

**Drosha regulates DNA:RNA hybrid formation in the proximity of DSBs**. The lack of detectable small RNAs arising from endogenous DSBs generated by the AsiSI endonuclease led us to examine other types of RNA species at DSBs. These could include long de novo synthesized RNA molecules arising after damage or pre-existing RNA transcripts still associated with the genomic region from previous or ongoing transcription events. Notably, Drosha has been previously documented to have functions in both transcriptional activation[36] and termination[20] and these activities may provide a clue to its roles in DNA damage-related RNA processing.

During transcription, DNA:RNA hybrid structures—so-called R-loops—are formed transiently and are ordinarily thought to predispose the DNA to damage[37]. Recently however, DNA:RNA hybrids were shown to be induced around DNA damage sites and to participate in the DNA repair process in yeast[38]. To determine if such a phenomenon is conserved in humans, we sought to visualize DNA:RNA hybrid formation around DSBs. Cells were transfected with an mCherry-tagged, catalytically inactive RNase H1 that specifically recognizes DNA:RNA hybrids[39]. Following DNA damage induced by laser microirradiation, rapid relocation of RNase H1 (Fig. 7a), but not the mCherry control (Supplementary Fig. 14A), to the laser track was observed. This suggests that DNA:RNA hybrids are formed upon DNA damage induction and that this occurs almost immediately following damage. Furthermore, depletion of both Dicer and Drosha resulted in a significant delay in the relocalisation of RNase H1 to the site of laser irradiation (Supplementary Fig. 16A), indicating that both

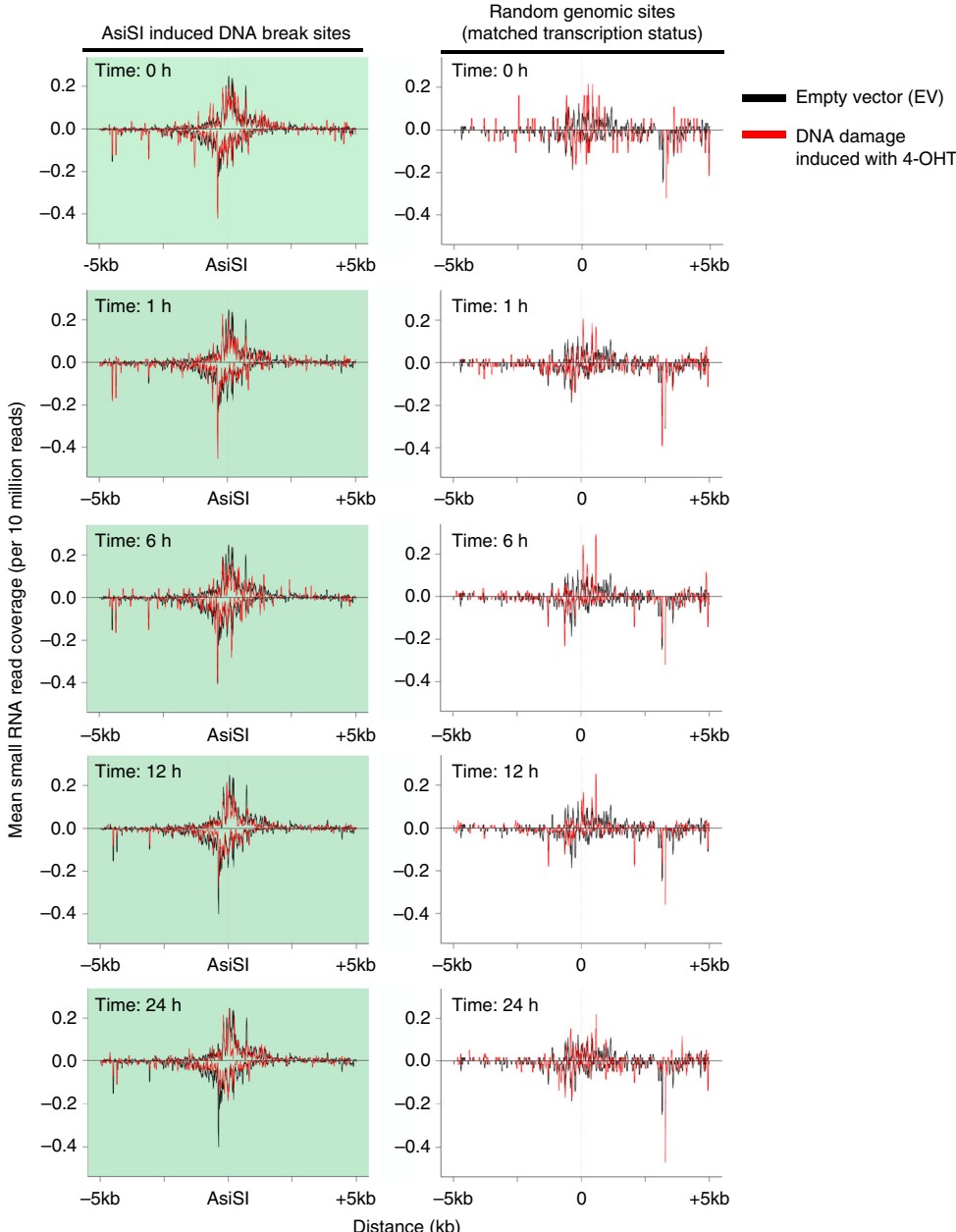

**Fig. 6** No significant enrichment of small RNA is found around 99 known AsiSI cut sites in the genome. Coverage profiles of 21–23 nt non-miRNA small RNAs that map to regions 5 kb on either side of AsiSI cut sites at different time points after induction of damage. Negative coverage refers to those reads mapping to the consensus genome minus strand. Left, 99 cut AsiSI sites (shaded with pale green). Right, same number of random genomic sites with similar transcriptional activity to the 99 AsiSI cut sites, as determined by TPM count and gene length from 4sU-Seq data on control cells (analysis described in Methods). These graphs are centred around a point (denoted as 0) at the same distance from the TSS as the matched AsiSI site. Each time point in the AsiSI transfected samples (red lines) is superimposed on reads from the EV control (black lines). As the EV control has never been exposed to the specific cuts produced by AsiSI, there is no potential for any small RNAs produced by that event to be present. Any reads present in the EV control can therefore be considered background

proteins have a role in the formation of DNA:RNA hybrids after DNA damage induction. To directly investigate the formation of DNA:RNA hybrids around break sites we utilized the S9.6 antibody, which specifically recognises DNA:RNA duplexes, for DNA:RNA-immunoprecipitation (DRIP)[40] in the inducible AsiSI endonuclease system. Remarkably, qPCR analysis of the hybrid-containing immunoprecipitated genomic fragments revealed a significant enrichment of these structures at damage sites that undergo repair by HR as well as NHEJ (Fig. 7b)[7]. This is specific to DSB sites because the level of R-loops at a highly transcribed locus (γ-actin) remains unchanged after damage induction. To

investigate the formation of R-loops on a genome-wide scale and determine if Drosha plays a role in this process, we conducted next generation paired-end sequencing of the DRIP genomic fragments (DRIP-Seq)[41]. We confirmed that DNA:RNA hybrids accumulate in the promoter and termination regions of genes in a transcriptional activity-dependent manner, as seen previously (Supplementary Fig. 14C)[42]. Consistent with our qPCR data, we observe a striking increase of DNA:RNA hybrid reads mapping to the proximity of AsiSI cut sites for break sites resolved by both HR and NHEJ[7], but not for uncut genomic loci (Fig. 7c). A sharp decrease of reads at the nucleotides corresponding directly to the

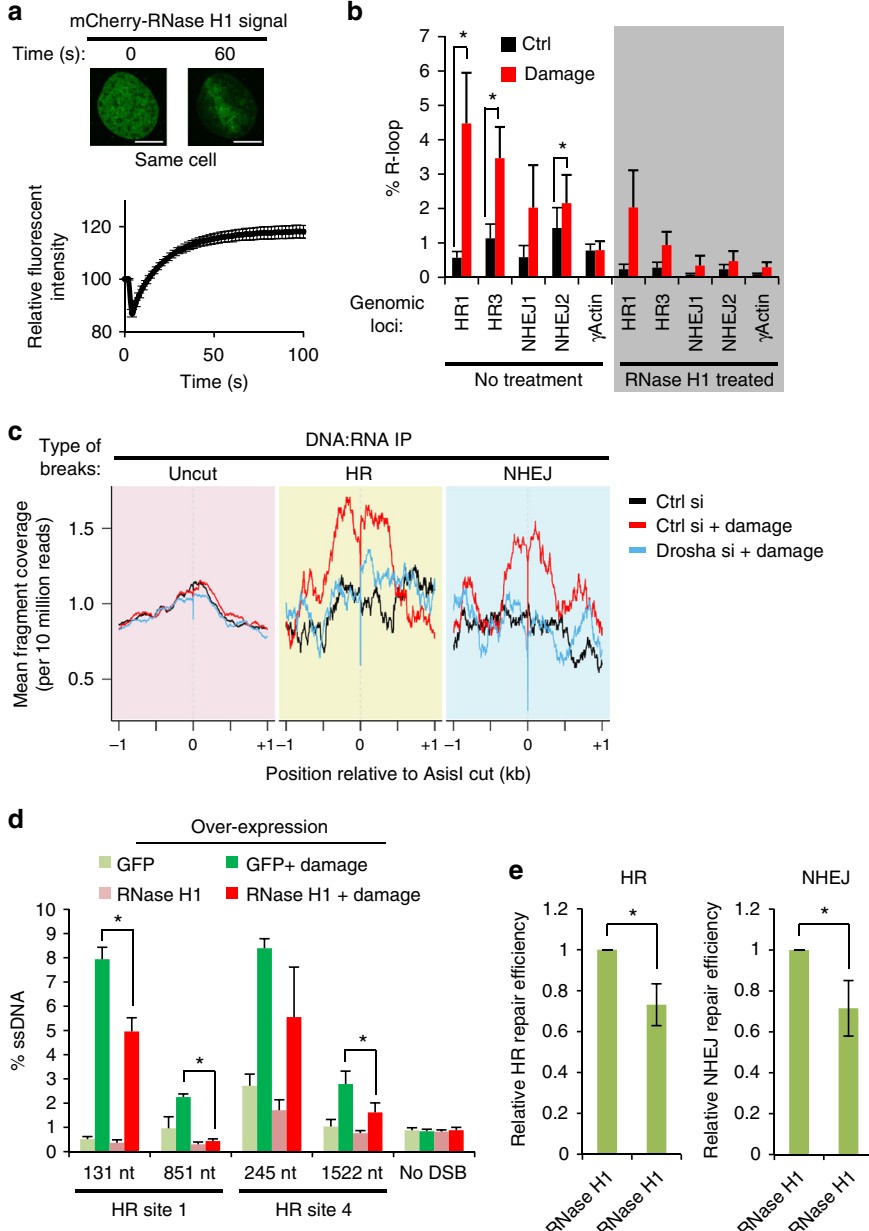

**Fig. 7** DNA:RNA hybrids form around DNA break sites to facilitate DNA repair in a Drosha-dependent manner. **a** Relocation of inactivated *E. coli* mCherry-RNase H1 D10R E48R to sites of laser-induced DNA damage. Representative fluorescence images, top. Scale bars, 10 μm. Bottom, graph showing quantitation of 168 cells over 3 replicates, error bars = SEM. **b** DNA:RNA hybrid IP (DRIP) followed by qPCR around HR and NHEJ DNA break sites, and control undamaged actin exon 5 locus after 2 h of damage induction. As a positive control, samples were treated in vitro with RNase H1 (shaded section). Error bars = SEM, Student's paired *T*-test, *$p \leq 0.05$ in 4 biological replicates. **c** DRIP-Seq was performed in conditions as in **b**. Graph shows enrichment of DNA:RNA hybrids around HR-repaired and NHEJ-repaired cut sites following DNA damage compared to sites documented to remain uncut following damage induction. **d** Over-expression of RNase H1 or a GFP control was followed by DNA resection assay as in Fig.5 a, b. *N* = 3, error bars = SEM, Student's 2-sample *T*-test, **$p \leq 0.01$. **e** RNase H1 was over-expressed in the HR (left) and NHEJ (right) repair reporter system cell lines (described in Fig. 4a) 6 h prior to I-SceI expression and GFP-positive cells were quantified as a measure of repair efficiency. *N* = 3 each, error bars = SD, Student's 2-sample *T*-test, *$p \leq 0.05$

2-nucleotide overhang left by the AsiSI enzyme confirm that these DNA:RNA hybrids form at actively damaged sites (Supplementary Fig. 14D). Since the DRIP-Seq library preparation includes a step in which 3′-overhang ssDNA fragments are exonucleolytically removed, this shows that the DNA:RNA hybrid forms prior to resection and the resolution of the break. Importantly, the depletion of Drosha abrogates the enrichment of DNA:RNA hybrids around break sites (Fig. 7c, Supplementary Fig. 14B, E, F). It is unclear if the observed hybrid structures are the result of de

novo synthesis following DNA damage or the increased interaction of pre-existing RNA molecules with their DNA template, but transcriptional activity of a locus prior to damage appears to predispose a site to the formation of DNA:RNA hybrids after damage (Supplementary Fig. 15A), while it does not seem to affect their cleavage efficiency (Supplementary Fig. 9). Curiously, observations at individual loci suggest transcriptional activity is not the sole determinant of damage-induced DNA:

RNA hybridization (Supplementary Fig. 15B), indicating other factors influence their formation.

As we have demonstrated that Drosha is required for proficient repair and for the formation or retention of DNA:RNA hybrids around DSBs, we hypothesized that DNA:RNA hybridization around DSBs is an important constituent of DNA repair. To test if the increase in DNA:RNA hybrids around DSBs facilitates the resolution of damage, we overexpressed active RNase H1, which specifically digests RNA in DNA:RNA hybrids[43], to reduce global occurrence of R-loops. This was followed by evaluation of resection efficiency at HR sites by qPCR. This showed that RNase H1 overexpression significantly reduced DNA resection, implying that the removal of R-loops impairs effective damage repair (Fig. 7d, Supplementary Fig. 16B, C). This is consistent with the recent discovery showing R-loop formation is required for DNA repair in yeast[38]. Overexpression of RNase H1 is currently the only way to investigate the consequences of R-loop formation in vivo[43] and we confirmed that this resulted in a decrease in both HR and NHEJ repair efficiency using the GFP reporter systems (Fig. 7e, Supplementary Fig. 16F). Combining this with depletion of Drosha did not have an additive effect on DNA resection, strongly suggesting that Drosha acts to promote the formation of R-loops (Supplementary Fig. 16D, E).

## Discussion

Results presented here show that Drosha plays a key early role in the DSB repair pathway. Its non-canonical activity was tested in the context of both ionizing radiation as well as an inducible restriction enzyme system that provides the opportunity to observe DNA repair in the endogenous genome at the nucleotide level[7,44].

Previous reports[14,23,45] have described a role for Drosha in DDR, but there were some discrepancies as to the precise step that it was acting upon. We have conducted a systematic screen of the successive recruitment of proteins to sites of damage, which revealed that Drosha is required at a very early step—around the time of RNF168/53BP1 recruitment (Figs. 1, 2, Supplementary Fig. 6) and that it affects both NHEJ and HR strands of the repair pathway (Fig. 4). Since 53BP1 and BRCA1 are key players in competing repair pathways and are considered antagonistic to each other[2,3], our observations demonstrate that the activity of Drosha in DDR goes well beyond a simple reduction in 53BP1 recruitment.

Intriguingly, BRCA1 has been shown to interact with Drosha and regulate miRNA biogenesis by virtue of its RNA-binding capacity[46]. Whilst we have ruled out the canonical miRNA pathway (Figs. 1, 4, Supplementary Fig. 6), it is tempting to hypothesize that BRCA1 and Drosha may be working in concert to orchestrate effective pathway choice and the ultimate resolution of the DSB.

Using the AsiSI endonuclease system we have globally examined the generation of small RNAs from the vicinity of endogenous genomic DSB regions. This intensive next generation sequencing approach failed to find evidence for this class of small RNAs at any of the endogenous cut sites (Fig. 6, Supplementary Fig. 11), even though it is clear that DSB repair pathways require both Drosha and Dicer in this system. These observations appear to be in conflict with previous reports where this novel class of small RNAs has been observed[14–16,32]. There are a number of possible explanations for this, most notably the distinct nature of the damage sites previously investigated. Our study focused on endogenous non-repetitive regions of the genome which have been well-annotated[7]. Previous studies have either looked in repetitive genomic regions[14,32], where accurate read alignment is problematic and mechanisms of repair may be distinct[47], or sites

artificially integrated into the genome[14,15,17]. In addition, recent findings in plants show that small RNA could only be detected from a reporter locus but not endogenous loci[35].

Because of the similar biochemical nature of RNA and DNA, one can envisage several mechanisms through which RNA could facilitate the repair process. This could be achieved by either playing an active role by interacting with complementary DNA at the break, acting as signaling molecules, or possibly templating repair, or passively by providing a platform for repair proteins[10,48,49].

We directly demonstrate for the first time in mammalian cells that DNA:RNA hybrids form around DSBs to facilitate repair, as was recently observed in S. pombe[38], and that this is Drosha-dependent either for their formation or retention (Fig. 7). We show that DNA:RNA hybrid formation is a very early event that precedes resection. Removal of DNA:RNA hybrids by RNase H1 overexpression results in impaired DNA damage repair (Fig. 7d, e). Whilst the presence of R-loops has traditionally been linked to predisposition to DNA damage[37], given our expanding understanding of the pervasiveness of transcription throughout the genome[50], it is clear that complementary RNA will be co-localized with multiple genomic loci upon damage. This exciting finding provides further evidence that RNA may be acting as a beacon of DNA damage, possibly recruiting repair machinery to the double strand break. Intriguingly, a recent publication characterized a resection-dependent end joining pathway in G1 phase of the cell cycle[51]. It has been speculated that in the absence of a sister chromatid, the error-free repair in this pathway could only occur in the presence of an RNA template[52], which would require hybridization to the DNA around the break site. The discovery of this novel repair pathway may go some way to explaining our observation that DNA:RNA hybrid formation around DSBs occurs at sites prone to both NHEJ and HR.

With regards to the role of Drosha in the formation or resolution of R-loops, much remains to be discovered. The recently described roles for Drosha in transcriptional activation[36] and termination[20] are interesting possible mechanisms through which this might be occurring, especially if the RNA component of the DNA:RNA hybrids is a damage-dependent transcription event. Alternatively, it may simply facilitate the connection between a protein, such as BRCA1[46], and one of its cofactors, such as DHX9 and DDX1, which have been demonstrated to be involved in R-loop unwinding[53,54]. It remains to be seen whether Drosha is directly involved in the Rad52-mediated RNA strand invasion, as reported in ref. [55]. This report suggested that Rad52 mediates DNA:RNA hybrid formation as a first step in an RNA-templated repair pathway.

Irrespective of the mechanisms finally determined, it is clear that Drosha and RNA play a central role in the resolution of DSBs. The emerging role for DNA:RNA hybrids appears to be highly conserved[38,48] and adds a new dimension to the repair process. While further work is required to understand the role of these structures in DNA repair, these data will facilitate mechanistic investigations to uncover their precise activities.

## Methods

**Cell line, cell culture, and transfection**. U2OS-HA-ER-AsiSI cells are described in ref. [44]. A549 (CCL-185) and U2OS (HTB-96) cells were obtained from ATCC. All cell lines are maintained in Dulbecco Modified Eagle's Medium (DMEM, GibCo) fortified with 10% Fetal bovine serum and 2 mM L-glutamine.

DNA was transfected using GeneJammer (Agilent), and GeneJuice (EMD Millipore). siRNA transfection was carried out using Dharmafect 1 or Dharmafect 2 (Thermo Fisher Scientific) for A549 cells and U2OS cells respectively. Non-targeting control siRNA (D-001810–03) and BRCA1 siRNA (J-003461–09) are from Dharmacon. Drosha siRNA sequence (CGAGUAGGCUUCGUGACUUdTdT) was documented in ref. [20]. 53BP1 siRNA sequence (GGACUCCAGUGUUGUCAUUdTdT) was documented in ref. [30]. The remainder of the siRNAs used in this study are obtained from Life Technologies:

TNRC6A (s26154), TNRC6B (s23060), TNRC6C (s33601), Dicer (s23754), and RNase H1 (s48356). Alternative siRNA used are Dicer (Dharmacon siGenome siRNA D-003483–03) and Drosha (Ambion s26490).

For the rescue experiment in U2OS-AsiSI cells, the siRNA resistant Drosha plasmid was transfected using the Nucleofactor kit V (Lonza, VCA-1003). siRNA transfection was carried out the following day using Dharmafect 2 (Thermo Fisher Scientific). Cells were harvested 48 h after the siRNA transfection. Experiments involving Drosha siRNA with RNase H1 overexpression were carried out in a similar manner using combination of Nucleofactor kit V transfection for the RNase H1 plasmid and Dharmafect 2 transfection for the siRNA.

All cell lines used were negative for mycoplasma contamination as tested for in-house (MRC Toxicology Unit, UK).

**DNA plasmid constructs**. The HA-ER-AsiSI plasmid is described in ref. [44]. HA-ER-AsiSI was cloned into the mammalian expression vector pCI-neo using the below primers to introduce SacII and SalI endonuclease sites into both donor and recipient sequences.

AsiSI F: TTG GTC CGC GGA ATT CAC CAT GGC ATA CCC
AsiSI R: AGCTGGTCGACTCACAACATC
pCI R: AACCACCGCGGATCGCTCGAGGCTAGCCTATAG
pCI F: TTG GTG TCG ACG GTT CCC AAT AGC TGA AGC GG

The inactive RNase H1-mCherry plasmid used to detect R-loop formation in laser microirradiaton studies was a gift from Patrick Calsou (Addgene plasmid #60367) as described in ref. [39]. Active mammalian GFP-tagged and FLAG-tagged RNase H1 for over-expression experiments was a gift from Thomas Tuschl (Addgene plasmid #65,784 and #65,782). HR and NHEJ reporter was a gift from Nickolai Barlev[29].

Firefly and Renilla reporter constructs were described in ref. [56]. Flag-Drosha plasmid was described in ref. [57] and was a gift from Shuo Gu (NIH, Frederick, USA). To produce an siRNA resistant version of Drosha plasmid, mutagenesis was performed using the following primer pair 5′- CTACAGTGGTTGGAACGAGTA GGCTTCGGGATCTATATGACAAATTTGAGGAGGAGTTGGG-3′; 5-CCTCCT CAAATTTGTCATATAGATCCCGAAGCCTACTCGTTCCAACCACTGTAGAA TCTCCC-3′.

**Western blotting**. Treated cells were harvested on ice with RIPA lysis buffer [400 mM NaCl, 1.5% Igepal CA-630 (Sigma-Aldrich), 0.5% Sodium deoxycholate, 0.1% sDS, 50 mM Tris pH 8.0, 1× complete protease inhibitor cocktail (Roche), 1× phosphatase inhibitor cocktail (J63907, Alfa Aesar), 15 μM MG-132 proteasome inhibitor (Tocris Bioscience)]. Samples were then sonicated for 5 min at high setting (Diagenode Bioruptor) in order to shear chromatin. The supernatant was collected after 15 min centrifugation (13,000 RCF, 4 °C). Western blotting was carried out on supernatants as described in ref. [58]. IR-Dye-labeled secondary antibodies (Li-COR Biosciences) were used at 1:15000 dilution. Blots were scanned with Li-COR Odyssey scanner (Li-COR Biosciences) and analysed using Image Studio software V2.1. Uncropped scans for main figure blots are shown in Supplementary Fig. 17.

**miRNA profiling by RT-qPCR**. A549 cells were transfected with siRNAs against Drosha, Dicer, TNRC6A-B, and a control for 48 h. Total RNA was collected following the Trizol protocol. The relative level of miR-21 was determined following the TaqMan Small RNA Assay protocol. Briefly, 100 ng total RNA was reverse transcribed in a 15 μl reaction using the TaqMan MicroRNA Reverse Transcription kit (Applied Biosystems, 4366597) and either hsa-miR-21 RT primer (TaqMan Assay ID 000397) or the control snRNA U6 RT primer (TaqMan Assay ID 001973). A no-template control reaction was carried out in parallel. After reverse transcription, cDNA was diluted 1:5 in water, and 5 μl of this was used in triplicate qPCR reaction with the appropriate assay probe supplied alongside the RT primer and TaqMan Fast Universal PCR master mix (Applied Biosystems, 4367846). The qPCR reaction was carried out on an Applied Biosystems 7500 Fast Real Time PCR system using the built-in TaqMan assay reaction settings. $\Delta C_T$ values were calculated by subtracting the mean paired U6 values from the mean miR-21 $C_T$ values for each condition; $\Delta\Delta C_T$ by subtracting the control siRNA $\Delta C_T$ from each knockdown sample. miR-21 abundances relative to U6 snRNA was calculated as $2^{-\Delta\Delta CT}$.

**Small RNA sequencing**. U2OS cells were transfected with pCI-HA-ER-AsiSI (pCI-AsiSI) or pCI-neo empty vector (EV) by electroporation using the Nucleofactor kit V (Lonza, VCA-1003) following the manufacturer's protocol. 18 h later, 300 nM 4-OHT or an equal volume DMSO was added. At 1, 6, 12, and 24 h after addition of 4-OHT, small RNA was harvested using Trizol with an additional phenol-chloroform purification and ethanol precipitation step, as described previously[58]. All precipitation steps were carried out overnight at −20 °C with the addition of glycogen to maximize small RNA recovery. Protein was collected in parallel to validate damage induction by western blotting.

Small RNA libraries for NGS were prepared following a protocol to increase efficiency and reduce biases in adapter ligation[59]. 600 ng total RNA was ligated to 3′ and 5′ adapters containing 4 random nucleotides at the ligating ends. Following 3′ adapter ligation, excess adapter is removed by RecJ exonuclease (Epicenter,

RJ411250) followed by purification of the RNA (RNA cleaning and concentration kit, Zymo, R1015) to further increase small RNA sequencing efficiency by reducing adapter-adapter ligations. Two biological repeats were sequenced on Illumina NextSeq500 1 × 75 bp high output mode generating 400 M single end reads.

The 3′ adapter sequence (TGGAATTCTCGGGTGCCAAGG) was removed from small RNA reads using cutadapt v1.11[60]. Cutadapt was also used to indiscriminately remove the degenerate 4 nt sequence from both sides of the adapter-trimmed reads. Trimmed reads were aligned to the human genome (GRCh38 from Ensembl; www.ensembl.org) using Bowtie2 v2.2.9[61]. Mapped primary alignments were further aligned to the human primary miRNA hairpin sequences (miRBase sequence database release 21, www.mirbase.org)[62] to detect all miRNA reads. Those reads that did not map to miRNA hairpins were further filtered using samtools v1.3.1[63] to remove other small RNA genes, including tRNA, snRNA, snoRNA, and scRNA as annotated by DASHR[64] and also annotated Y RNA and sRNA with ensembl reference release 85. The final remaining reads were considered non-miRNA small RNAs and only these were used for further analyses.

21–23 nt reads were extracted as previously reported damage-dependent small RNAs were shown to be this size[14]. Reads in the 19-20 nt or 24–26 nt size groups were also analysed. Those that mapped to the regions 5 kb either side of every AsiSI recognition site in chromosomes 1–22 & X ($N = 1220$) were counted using HTSeq-count v0.6.1p1[65]. Differential expression was performed using DESeq2 1.12.4[66]. Of the potential 1220 AsiSI recognition sites, around 100 are accessible for cutting in the intact cell[67]. A list of 99 of the known cut AsiSI sites[7] was kindly supplied from the Legube lab, University of Toulouse, France. As most AsiSI recognition motifs in the genome are therefore uncut, no changes to the small RNA levels around these should be seen. As such, the DESeq2 model should observe even minute significant changes at those sites that are cut. To detect small RNA as a result of AsiSI cutting, the DESeq2 model was designed to compare the 0 h time point and all damage time points. To graphically represent these results, a volcano plot was generated by plotting −log10(unadjusted p-value) against log2(fold change over time). As all adjusted p-values (false discovery rates) were close to 1, the unadjusted p-values were used for plotting to show variability.

To visualize these counts around break sites, read coverage was calculated in 10 bp windows 5 kb either side of each of the 99 AsiSI cut sites using bedtools v2.26.0[68] and normalized to total non-miRNA small read count and plotted in R using ggplot2 v2.1.0. A set of control sites with similar transcriptional activity (as determined from 4sU incorporation; see below) was also plotted to compare against the AsiSI cut sites.

**5′-Triphosphate small RNA sequencing**. To prepare any small RNA species with 5′-triphosphate moieties, total RNA was first treated with 1 U Terminator 5′-phosphate-dependent exonuclease (epicenter, TER51020) for 30 min at 30 °C in the supplied buffer. This degrades all RNA with 5′-monophosphates (miRNA, rRNA) and leaves 5′-di/triphosphate and capped RNA intact. RNA was acid phenol: chloroform extracted and ethanol precipitated as before. The Terminator-treated samples were then treated with 5′-polyphosphatase (epicenter, RP8092H) for 30 min at 37 °C to remove the γ-phosphates and β-phosphates, leaving them as 5′-monophosphates. RNA was again extracted by acid phenol-chloroform and ethanol precipitated. These samples were then used for small RNA sequencing library preparation as above.

**Transcriptional activity determination by 4sU incorporation**. Metabolic labeling of newly transcribing RNAs was carried out as in ref. [69]. In short, U2OS incubated with 0.5 mM 4-thiouridine (4sU) (Sigma, T4509) for 1 h. RNA was harvested by Trizol. Eighty microgram of total RNA was then subjected to a biotinylation reaction (1 mg/ml EZ-link HPDP-biotin, Thermo Scientific #21341) for 90 min at RT followed by chloroform extraction and RNA precipitation. The nascent RNA species containing biotinylated 4sU nucleotides were then specifically separated from non-4sU-containing RNAs by streptavidin-coated magnetic bead pulldown. Following several washes, 4sU-containing RNA was eluted with 100 mM DTT and ethanol precipitated.

4sU-enriched RNA was depleted of ribosomal RNA (Ribo-Zero, Illumina) and subjected to Illumina TruSeq Stranded RNA library preparation omitting the poly-A purification step, carried out by the DNA Sequencing Facility, Department of Biochemistry, University of Cambridge. Stranded libraries were sequenced on the NextSeq500 2 × 75 bp high-output mode.

Paired fastq files were aligned to the human transcriptome (GRCh38.85) using the pseudoaligner kallisto v0.43.0[70]. The transcripts per million (TPM) values as determined by the quantification algorithm were then used to randomly select genes with similar lengths and TPMs to each AsiSI cut gene. As all RNAs sequenced were enriched from a 4sU incorporation pulse, genes matching these criteria were considered to have a similar transcriptional activity to those AsiSI cut genes and used as a control for small RNA coverage plots.

**DNA:RNA IP (DRIP)**. The DRIP protocol was modified from ref. [40]. Briefly, U2OS-HA-ER-AsiSI cells were washed and lysed in cytoplasmic lysis buffer (10 mM HEPES pH7.9, 10 mM KCl₂, 1.5 mM MgCl₂, 0.34 M sucrose, 0.1% TritonX-100, 10% glycerol, 1 mM DTT). The nuclear pellet was obtained by centrifugation at 1500 RCF. The chromatin fraction was obtained by incubating the nuclear pellet with a buffer containing 50 mM Tris pH 8.0, 5 mM EDTA, 1% SDS and Proteinase

K at 50 °C for 2 h, and centrifugation after addition of KOAc to the final concentration of 1 M. Genomic DNA containing R-loops was then precipitated overnight from the chromatin fraction by the addition of ethanol. Genomic DNA was washed with 70% ethanol, re-suspended in water, quantified, and then digested with AsiSI for 2 h. This enzymatic digestion was carried out to ensure the genomic fragments of damaged and undamaged samples are of comparable length for deep sequencing purposes. The DNA was ethanol precipitated again, re-suspended in 400 µl IP buffer (16.7 mM Tris pH 8.0, 1.2 mM EDTA, 167 mM NaCl, 0.01% SDS, 1.1% Triton X-100) and sonicated for 10 min (Diagenode Bioruptor, Medium setting, 30 s on/30 s off interval). The length of resulting DNA fragments was inspected on 1.5% agarose gel and verified to be between 300 and 1000 nt. Half of the genomic DNA was treated twice with RNase H1 (NEB, M0297S, 3 units per µg DNA input, 37 °C 3 h) as negative control. For DRIP, 1 µg S9.6 antibody (Kerafast, ENH001) conjugated with ChIP grade protein A/G magnetic beads (Thermo Scientific, #26162) was used for every 2 µg genomic DNA input. The resulting immunoprecipitation was carried out at 4 °C. The beads were washed once in IP buffer, once with IP buffer with 500 mM NaCl, and once with LiCl wash buffer (10 mM Tris pH 8.0, 250 mM LiCl, 1 mM EDTA, 1% NP-40), and once with TE with 50 mM LiCl. The magnetic beads and their respective input genomic DNA, were re-suspended in 400 µl nuclease-free water, and treated with RNASE 1 at 37 °C (Invitrogen, AM2295). The samples were then subjected to 2 rounds of Proteinase K treatment at 55 °C, and DNA extracted with phenol:chloroform:iso-amylalcohol (pH 8.0, Sigma P2069), followed by ethanol precipitation. The amount of immunoprecipitated DNA:RNA hybrid around a genomic region was then assessed by qPCR. The sequences of qPCR primers are shown in Supplementary Table 3.

**DRIP-Seq.** One nanogram of DRIP-ed genomic fragments were prepared for NGS using the NEBNext Ultra II kit and multiplex oligos (NEB, #E7645S & E7335S) following the manufacturer's protocol using 8 PCR amplification cycles and an additional 0.85× volume clean-up using the supplied sample purification beads. Libraries were sequenced on the Illumina NextSeq500 2 × 75 bp high-output mode generating 800 M paired-end reads (DNA Sequencing Facility, Department of Biochemistry, University of Cambridge).

Sequenced paired fastq files were aligned to the human genome (GRCh38) using Bowtie2 2.2.9[61]. Mapped paired alignments were then extended to get the full sequenced fragment by taking the 5′ end of the most 5′ mapping read and the 3′ end of the most 3′ mapping mate to create a BED file used for further analyses. Interchromosomal pairs and extremely long (>2500 bp) intrachromosomal pairs were ignored for these analyses. The DRIP-ed DNA fragment coverage was plotted around the 99 known cut AsiSI sites. These were split into groups based on their preferential repair[7]; a list of 25 sites confidently assigned as being repaired by either HR, NHEJ, or those 49 remaining as unspecified. The log2 ratio of fragment counts in the damaged condition to those in the undamaged condition for control and Drosha siRNAs were also plotted according to repair type to show enrichment or depletion of DNA:RNA immunoprecipitated genomic regions.

**Chromatin immunoprecipitation (ChIP).** U2OS-HA-ER-AsiSI cells were treated with 300 nM 4-OHT to induce DSBs. The cells were harvested by trypsination, counted, re-suspended in DMEM, and crosslinked in 1% formaldehyde for 15 min. Protein crosslinking was quenched by the addition of glycine to 125 mM final concentration for 5 min. 7.5 million cells were washed with ice cold PBS twice then lysed in cytoplasmic fractionation buffer (10 mM HEPES pH 7.9, 10 mM KCl₂, 1.5 mM MgCl₂, 0.34 M sucrose, 0.1% TritonX-100, 10% glycerol, 1 mM DTT). Nuclear pellet was recovered by centrifugation at 1500 RCF, and lysed in 375 µl nuclear lysis buffer (50 mM Tris pH 8.0, 5 mM EDTA, 1% SDS). Sonication was carried out using Bandelin Sonopuls HD2070 sonicator, at 50% amplitude, 10 cycles of 10 s on/off pulse. The length of resulting DNA fragment was inspected on a 1.5% agarose gel and verified to be between 300 and 1000 nt. ChIP protocol was carried out as per ref. [7]. Four microgram of antibodies (described in Supplementary Table 1) pre-conjugated with ChIP grade protein A/G magnetic beads (Thermo Scientific, #26162) were used per 100 µl input and immunoprecipitated for 2 h. All antibodies were validated in ChIP by the providers. Immunoprecipitated DNA and respective input DNA were analysed by qPCR. Primer sequences of qPCR are provided in Supplementary Table 2.

**Microscopy and DNA damage foci studies.** Preparation of slides for DDR immunofluorescence foci studies was carried out as described in ref. [71]. Briefly, A549 or U2OS-HA-ER-AsiSI cells were pre-seeded on glass cover slips, subjected to IR irradiation (Xstrahl RS320) /HA-ER-AsiSI induced DNA damage, and allowed to recover for 30 mins—6 h. Cells were then washed once with PBS, and pre-extracted with CSK buffer (100 mM NaCl, 300 mM sucrose, 3 mM MgCl₂, 10 mM PIPES pH 6.8, 10 mM β-glycerol phosphate, 50 mM NaF, 1 mM EDTA, 1 mM EGTA, 5 mM Sodium orthovanadate, 0.5% Triton X-100). Cells were then washed once with CSK buffer and fixed with 4% paraformaldehyde on ice for 20 mins. Samples were then washed thrice with 0.1% TBS-Tween, blocked with 10% goat serum for 1 h, washed twice with 0.1% TBS-Tween and incubated with primary antibody overnight in 4 °C (see Supplementary Table 1). The samples are then washed and incubated with secondary antibodies (Alexa Fluor 488/546, Invitrogen)

in 1% goat serum for 1 h at room temperature. Samples were then washed with 0.1% TBS-Tween and mounted with Vectashield with DAPI (Vector laboratories, H1200). The samples are then double-blinded, and images were obtained with Carl Zeiss LSM501 confocal microscope, using 60× objectives. For consistency, the laser settings between each biological repeat are the same. Images were then processed, un-blinded, and counted in an unbiased way using the FindFoci ImageJ plugin[72]. Each experiment was conducted in three independent biological repeats, counting at least 60 cells per condition for each repeat. All images of a given experiment set were analysed under the same parameters. The number of foci per cell was then plotted as both raw data points and also as a violin plot with medians. Statistical testing was carried out using Mann–Whitney non-parametric test to compare a singular condition against control; in the case of multiple conditions, a Dunn's test with Bonferroni corrections for multiple comparisons was performed.

**Single-stranded DNA resection assay.** Single-stranded DNA resection assay was carried out as per ref. [30]. Briefly, for Drosha knockdown experiments, U2OS-HA-ER-AsiSI cells are plated overnight, Control or Drosha siRNA were transfected for 48 h using Dharmafect 2 (Thermo Scientific) before subjected to 4 h incubation in 300 nM 4-OHT. Similarly, GFP-RNase H1 over-expression are carried out by transfecting respective plasmids for 48 h using GeneJuice (Agilent Technologies) before subjected to 4-OHT induced DNA damage. Genomic DNA was extracted using DNeasy Blood and Tissue kit (#69504, Qiagen). Every 500 ng genomic DNA was treated with 5 units of RNase H1 (NEB M0297) at 37 °C for 15 min. As described in Fig. 3c, in vitro restriction digestion is required to assay for the presence of ssDNA around break sites. Depending on the DNA sequence around individual break site, different resection enzymes are used (see Supplementary Table 2 for detail). Two hundred nanogram of samples were digested with 16 units of respective restriction enzymes at 37 °C for 12 h (BsrGI-HF, BamHI-HF, HindIII-HF, BanI, and PstI-HF; NEB). The percentage of ssDNA generated by DNA resection was determined by qPCR as described in ref. [31]. A list of qPCR primers used in this study can be found in Supplementary Table 2. $\Delta C_T$ is defined as the difference in average cycles between a given digested sample and its undigested counterpart. To calculate % ssDNA the following equation was applied: % ssDNA $= 1/[2^{\wedge}(\Delta C_T - 1) + 0.5] * 100$.

**Ligation-mediated DNA cleavage assay.** The full procedure of the cleavage assay was described in ref. [7,73]. Briefly, biotinylated dsDNA oligonucleotide containing an AsiSI site 3′ overhang was ligated in vitro to genomic DNA after in vivo induction of AsiSI breaks. eIF4A2 plasmid linearized with AsiSI in vitro was spiked into the reaction mix to serve as positive control. After ligation, DNA was fragmented with EcoRI for 2 h, and incubated with streptavidin beads (M-280, Thermo Fisher Scientific) for 18 h at 4 °C. The beads were washed once in IP buffer, once with IP buffer with 500 mM NaCl, and once with LiCl wash buffer (10 mM Tris pH 8.0, 250 mM LiCl, 1 mM EDTA, 1% NP-40), and once with TE with 50 mM LiCl. The magnetic beads and their respective input genomic DNA were re-suspended in 100 µl nuclease-free water and digested with HindIII at 37 °C for 4 h. After phenol/chloroform purification and precipitation, DNA was resuspended in nuclease free water and assayed by qPCR. qPCR primers used are listed in Supplementary Table 2.

**Luciferase activity assay.** Luciferase activity assay was carried out as per described in ref. [56]. The pRL renilla luciferase reporter plasmids containing 0/2× let-7 miRNA target sites, and control pGL3 firefly luciferase reporter plasmids are also described in detail in ref. [56].

**Laser microirradiation.** For GFP-53BP1 laser microirradiation, GFP-53BP1 expressing U2OS cell line was carried out as described previously[28,74]. U2OS cells stably expressing eGFP-53BP1 were reverse transfected with siRNA, 24 h later, cells were pre-sensitized with 10 µM BrdU for a further 24 h prior UVA laser micro-irradiation. Experimental procedure is detailed in ref. [28,74].

To monitor DNA:RNA hybridization in real time following DNA damage, a catalytically inactive but R-loop binding capable RNase H1 tagged with mCherry was transfected into U2OS cells seeded into Lab-Tek chambered coverglass wells (Thermo Scientific #155383). Twenty-four hours later cells were incubated with 20 µM Hoescht 33342 (Thermo Scientific #62249) for 10 min to pre-sensitize to laser damage. Cells were maintained at 37 °C and 5% CO₂ for the duration of the experiment in a stage top environmental control system (okolab). Image acquisition was performed using a spinning disk inverted microscope system (3i). Using 3i Slidebook software, laser track lines were drawn across cell nuclei and microirradiation induced using a 532 nm pulsed laser (3i Ablate! model 3iL13). The 561 nm channel was imaged every 500 ms monitoring the intensity along the laser stripe. This was normalized to the background intensity of a second undamaged stripe within the nucleus and the first time point set to 1.

**Flow cytometry.** For cell cycle analysis, propidium iodide staining of DNA was carried out as previously described[58].

For analysis of cell survival, an Annexin V/DRAQ7 assay was used. U2OS cells were treated with 1 µg/ml bleomycin (Cambridge Bioscience) or vehicle for 24 h and the media, PBS-washed and adherent cells were harvested and incubated

at 37 °C for 20 min in DMEM. After centrifugation (300xg, 5 min RT) the supernatant was removed and the cells resuspended in Annexin buffer (10 nM HEPES pH 7.4, 150 nM NaCl, 2.5 mM CaCl₂) containing Annexin V-FITC (1/2500, kind gift from Marion MacFarlane, MRC Toxicology Unit, Leicester). Tubes were incubated for 30 min at room temperature before addition of DRAQ7 (375 nM final; abcam, ab109202). Samples were immediately run on FACS Canto (BD Bioscience) and the fluorescence intensity of FITC channel and APC channel used to gate populations indicating non-apoptotic, early, and late apoptotic cells and necrotic cells.

Repair reporter cell lines were generated as described in ref. [29]. Briefly, U2OS cells were electroporated with NheI-linearized HR or NHEJ reporter plasmid. Twenty-four hours following transfection, chromosomally integrated reporter cells were isolated by selection with 1 mg/ml G418 (Sigma, A1720) for 1 week. Individual cells were then plated in a 96-well plate and supplemented with 1:1 conditioned U2OS media to fresh media with 1 mg/ml G418 and progressively grown until sufficient cell numbers were reached. Cell lines were validated for correct integration of the repair reporter by transfecting the I-SceI overexpression plasmid for 48 h and measuring GFP induction by flow cytometry (see below). The cell lines with the highest GFP positive readings were then further validated for HR or NHEJ repair by depleting BRCA1 or 53BP1, respectively by siRNA and observing ablation of the GFP signal. The cell line with the greatest response to these criteria was then used in this study.

For investigating the effects of the miRNA pathway components on repair, the reporter cell line in 6-well format was transfected with 30 nM siRNAs against given proteins for 48 h followed by transfection with 2 µg I-SceI plasmid for a further 48 h. Media was replaced the day after each transfection. Cells were trypsinised and resuspended in 800 µl media. Control cells with no transfection and with GFP overexpression were plated in parallel and used to optimize gating conditions on a FACSCanto II flow cytometer (BD) controlled by FACSDiva software (v.8.0.1, BD). At least $3 \times 10^4$ cells were counted per condition, measuring cells as either GFP positive or negative. Raw numbers were exported and used to plot HR/NHEJ repair efficiency, which is defined here as the proportion of GFP positive cells per condition normalized to the control siRNA condition in each biological replicate. Statistical significance was determined using the Wilcoxon signed rank nonparametric test comparing the normalized values to the normalized control value of 1.0.

**Statistical analysis.** Unless stated otherwise, all error bars represent mean ± 1 SD of the experiment set, with numbers of repeats N are indicated in the figure legends. Statistical testing was performed using the Student's T-test, the Wilcoxon signed-rank test (for repair reporter assay, comparison of conditions to the normalized control value of 1), Mann–Whitney test (for non-parametric testing of the number of foci per cell, between control and a single condition), or Dunn's test with Bonferroni corrections for multiple comparisons (where Mann–Whitney is not appropriate for multiple conditions). Each test used is indicated in figure legends and in all cases, $*p \le 0.05$, $**p \le 0.01$, $***p \le 0.001$.

**Data availability**. All high-throughput sequencing data has been submitted to the NCBI Gene Expression Omnibus (GEO) under accession GSE97648. For analyses, workflows are described in the appropriate Methods sections and were performed in the command line (Ubuntu 14.04) or the R environment. Software was maintained at the latest stable versions. The version used at the time of analyses is reported in the above Methods.

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

## Acknowledgements

This work was funded by Medical Research Council (MRC). W.-T.L. was partially supported by an MRC Centenary Award. A.W. is funded by the BBSRC. The authors would like to thank Gaelle Legube (University of Toulouse, France) for providing the U2OS-AsiSI cell line and the AsiSI overexpression plasmid, advice, and assistance in data analysis.

## Author contributions

W.-T.L. and B.R.H. designed and performed most of the experiments, analysed data and contributed to writing the paper. R.A.B. and F.Z.W. performed 53BP1 laser tracking experiments. A.S.B. conducted several IF experiments. G.L.S and M.M. performed apoptosis assays. E.M.S analysed data and contributed to writing the paper. M.B. and A.W. coordinated the project, designed experiments, analysed data and wrote the paper.

## Additional information

**Competing interests:** The authors declare no competing financial interests.

