## [Peer Review File · Nature Communications]

Reviewers' Comments:

Reviewer #1:

Remarks to the Author:

The manuscript by Lu et al. reports that Drosha and Dicer are required for the efficient recruitment of repair factors to DSBs, but their effects are independent from their function in the classical miRNA pathway. The authors were unable to detect DNA damage induced small RNAs around DSBs and claim that the previously reported DSB induced small RNAs are likely the consequence of specific DSB systems used in those studies. The authors suggest a role for Drosha (and probably Dicer?) in the formation of DNA:RNA hybrids around DSBs.

Overall, the topic of the manuscript is very interesting and highly significant. An increasing body of evidence suggests that RNA and RNA processing enzymes are involved in DDR, but their role is not clear. However, in my opinion, the manuscript fails to provide solid evidence to back up most of its novel claims, while some other data has been previously published.

The role of Dicer and Drosha in DDR was previously reported in various organisms. Similarly, DNA:RNA hybrids were detected around DSBs, not only in *S. pombe*, but also in human cells (Britton et al. (2014) *Nucleic Acids Res.* 42, 9047–9062). The involvement of Drosha in this process is not very convincing (see major concerns below) and the authors did not suggest a feasible model to explain how Drosha would influence R-loop formation. The experimental evidence is weak to convincingly support the claim that there is no increase of small RNAs around DSB sites.

Major concerns:

There are several reports in various organisms describing the DNA damage dependent appearance of small RNAs around DSBs and these studies claim that these small RNAs are required for efficient DDR. One of the centrepieces of the study by Lu et al is the small RNA sequencing of U2OS cells before and after AsiSI induced DSBs at various time points. In these experiments, the authors could not identify DNA damage induced small RNAs around the DSBs. It is always delicate to prove the lack of something, because it can be easily argued that the sensitivity of the method is not good enough to detect – in this case the DSB induced small RNAs. However, this manuscript did not attempt to provide evidence that the experimental setup has similar sensitivity to previous studies which were able to detect small RNAs around DSB sites. There are several potential pitfalls during RNA isolation, library preparation and processing of the data which could lead to the loss of low abundant small RNAs. An even more serious problem is that the authors did not provide any data of the cleavage efficiency at individual AsiSI sites. Is an individual site cleaved in 10% of the cell population or 1% or only 0.1%? In this experiment, U2OS cells were transfected with the AsiSI plasmid and induced up to 24h. A WB in Fig S8B shows somewhat elevated pATM levels after 1 h which does not change after further induction, which is surprising. There is no data to evaluate the efficiency of the cleavage. The small RNA sequencing result in Fig 6 is also very hard to evaluate. What is the interpretation of the small RNA peak around the AsiSI cleavage sites at every time point, including the uninduced (0h) sample? Would these “background” small RNA peaks around the cleavage sites allow the detection of a lower abundant RNA and would this be a significant change in the statistical analysis?

Overall, after reading this manuscript, I am not more or less convinced about the existence or absence of small RNAs around DSBs than I was before.

The lack of convincing controls that show unaffected cleavage efficiency between Ctr and Drosha knock-down cells is also disturbing throughout the manuscript. The observed decrease in resection and/or DNA:RNA hybrid formation in Drosha knock-down cells could also be explained with decreased cleavage efficiency or slower induction kinetics in these cells, and as such, could be an indirect effect of the Drosha knock-down. This must be rigorously controlled, especially because the observed changes between Ctr and Drosha si are rather modest. The TNRC6A-B knock-down suggests that the Drosha effect is likely not mediated by the classical miRNA pathway, however TNRC6C could potentially also mediate miRNA effect.

Another issue is that the AsiSI system monitors only a special subset of genomic locations, namely those that can be accessed by the enzyme, which is less than 10% of the AsiSI sites. Why are 90% not accessible? Since AsiSI is sensitive to methylation, the accessible sites must be unmethylated. Would the majority of the sensitive sites be located in CpG islands of active promoters? (Could this explain the detected small RNA peak around the cleavage sites?) Or must these sites be at nucleosome free regions? Either way, the sensitive sites represent only a special subset of chromatin environment and we don't know if this has any consequence for the conclusions. While there is no perfect system for DSB induction, the limitations of each system have to be considered and clearly discussed.

Reviewer #2:

Remarks to the Author:

In their manuscript entitled: 'Drosha drives the formation of R-loops around DNA break sites to facilitate DNA damage repair.' Lu and co-workers study the roles of Drosha and Dicer in the repair of DNA double strand breaks.

Increasingly, we realize that RNA, and RNA processing enzymes, plays a role in the repair of DNA. However, much of how this is regulated remains unclear. In this manuscript a number of experiments are performed which re-inforce the role of RNA processing enzymes in DNA repair. siRNA-mediated depletion of Drosha or Dicer interferes with the timely accumulation of 53BP1, BRCA1 and Rad51 at IR-induced foci. Using GFP-53BP1 in combination with laser stripes, a delayed recruitment to DNA damage is confirmed. The requirement for Drosha or Dicer appears restricted to HR, as GFP-based repair assays show a strong defect in GFP conversion. Further, Drosha appears required for DNA end-resection. Mechanistically, Drosha seems to be involved in the generation of R-loops at sites of DNA damage, which were recently shown to be involved in DNA repair.

Overall, I find that the initial phenotypes are strong but descriptive (53BP1 recruitment, GFP conversion, end resection), but when it reaches the stage what Drosha does at the molecular level (DNA-RNA structures), I am less convinced. Much of the data is descriptive in nature, and to my opinion does not extend mechanistic insight sufficiently to warrant publication

Specific comments:

- How do Drosha and Dicer function in establishing R-loops at DSB sites? Is enzymatic activity required? Consequently, how does Drosha or Dicer inactivation interfere with DDR signalling at the level of 53BP1/BRCA1. Is R-loop formation required for ubiquitin signalling affected, for instance involving RNF8/RNF68?

-R-loop formation as investigated by RNase overexpression is shown to be involved in DNA break processing, confirming previous work. To extend beyond this finding, interference with R-loop (RNase overexpression) should also be studied in the context of Drosha/Dicer inactivation to see if these effects are epistatic

-To me it is a bit unclear whether to authors claim whether the effects are towards HR, NHEJ or both. To show differential effects on HR, it would be strong if control probes for NHEJ/SSA should be included (eg EJ5-GFP and SA-GFP). How do the authors explain that Drosha inactivation interferes with HR, but that R-loop formation happens at HR and NHEJ sites?

-I find the term 'DDR positive cells' misleading. It would be better to indicate absolute numbers of foci per cell in Figures 1,2 and 3.

-Figure 3: the kinetics of 53BP1 recruitment are different between foci and laser stripe. It seems that in the kinetic analysis

-Figure 5C: the results of the ChIP are displayed as relative values to a miRNA site. This way of displaying data makes it difficult to interpret results. Is there an effect of DNA damage of MiR-122 binding? Plotting data for the individual sites would be better.

- What is the consequence of Drosha or Dicer inactivation for cells? Does it lead to more mutations, translocations, or sensitivity to DNA damaging agents?

- Page 3/line 76: 'This confirmed that...' this is overstated. Should say that it is likely not linked to the canonical function.

-page 4/line 116: paragraph title is weird: is it required for HR? or proficient HR? please change.

Reviewer #3:

Remarks to the Author:

The study of Lu et al addresses several important and yet unresolved issues of DSB signalling and repair.

In brief, the authors address the ill-defined role of the RNase III enzymes Dicer and Drosha in DSB repair. They focus primarily on Drosha as they observe greater phenotypes upon abrogation of this enzyme. Interestingly, as part of their work they address the formation of small non-coding RNAs (so-called, damage-inducible or diRNAs) at 99 AsiS1 cut sites across the. Such damage-inducible small RNAs were previously reported by the d'Adda di Fagagna group at a non-native locus composed of repeat sequences either side of an Sce1 cut site. Despite their impressively rigorous analysis they find no evidence for small RNA produced at accessible AsiS1 sites (note my minor caveat to this statement below). Rather Lu et al report that induced RNA:DNA hybrids (R-loops), not diRNAs, are a general feature of DSBs. Their data indicates that Drosha is required for the formation of DSB-induced R-loops at genomic regions cut by the AsiS1 endonuclease and that these DSB-induced R-loops are required for effective resection of DSBs.

The specific molecular mechanism of how Drosha regulates the production of DSB-induced R-loops is not addressed in this study. However, subject to satisfactorily addressing my concerns below, and the influence this paper is likely to have on the field, this reviewer supports publication of this work in Nature Communications.

Specific comments:

The authors should make clear in the text whether they have used independent siRNAs to knockdown Dicer and Drosha. Only a single siRNA against each was indicated in the Material section. Also, a cDNA rescue of one representative phenotype resulting from Drosha knockdown should be clearly referenced in the text and shown in the supplementary materials. This will address the possibility of any misleading off target effects of the siRNAs used.

Some of the later experiments should be repeated with Dicer KD and some Dicer/Drosha double KD experiments should be included in an attempt to establish the epistatic relationship between both RNase III enzymes.

Fig 1A Dicer/Drosha KD cells have a mixture of bright and weak foci, whereas in control cells this phenotype is not as noticeable. Quantify bright and weak foci under all conditions. Does this mean that the Dicer/Drosha effect is specific to some foci but not others? Is this effect seen at other time points post IR. This is the key scene setting experiment of the paper. Therefore, DDR kinetics

should be investigated.

Not sure the % DDR positive cells is the ideal way to quantify this data. Count total foci or, better still, measure average focal intensity. However, the latter approach would need to be cognisant of the two classes of foci, assuming this is a real effect at multiple time points, as referred to above.

Fig 2. Title is an overstatement. While 53BP1 recruitment to DSBs is involved in regulation of repair pathway choice, a direct role in NHEJ is less clear. Technically it is correct to say "key events" but this could be over-interpreted by readers. A suggestion is to re-label as: "Drosha is required for focal recruitment of DDR factors downstream of MDC1".

Lines 137/138 of main text. Please check Legube paper (ref 7). From recollection these authors used presence or absence of RAD51 to indicate whether HR or NHEJ pathways were used at individual AsiS1 cut sites. Again, from recollection, XRCC4 was present at both categories of AsiS1 cut site.

Fig S2. Do Drosha KD cells really have more intense ATMph foci? Show quantification of this experiment. Why is there such a green background signal in the ATM-S1981ph panels.

Fig S7. As Dicer and Drosha are required to generate miRNAs, why does their KD have no effect on Let-7-dependent miRNA mediated repression? Please clarify.

Line 157 replace "profound" with "significant".

Fig 6. The small RNA sequencing is very nice and appears to this reviewer to be very rigorously performed. However, the analysis appears to be limited to "21-23 nt reads" based on Francia et al. 2012. Assuming that they exist, diRNAs might have a different size range. Therefore, the authors should examine other size ranges within their data sets, e.g. 19-20 nt and 24-26 nt.

Fig 7A. Investigate mCherry-RNase H1 recruitment upon depletion of Drosha/Dicer.

Fig 7C. Very nice DNA:RNA hybrid DRIP-seq data in human cells that is supported by recently published observations in yeast (ref 35). Regarding the issue of de novo transcription, have the authors considered using RNA pol II inhibitors? I appreciate the expense of these experiment, so I do not consider this to be essential for publication.

Fig 7D. Also a very nice preliminary experiment consistent with a positive role for R-loops in repair by HR. Enforced removal of R-loops by expression of RNase H1 abrogates resection and again consistent with the recent yeast data. Perhaps defective repair in the Jasin assay following expression of RNase H1 could be used to independently support the data with AsiS1 sites.

Label for Y axis missing from Fig S7A & B.

Reviewer 1

Overall comments:

A) The reviewer states that DNA:RNA hybrids have been detected around DNA DSBs in human cells, previously (Britton et al, 2014).

Our study shows for the first time these DNA:RNA structures at nucleotide resolution and, importantly, that:

- they form before the DNA is resected,
- their generation is Drosha dependent,
- they occur at both HR and NHEJ prone sites,
- they occur in the immediate proximity of the break site,
- their removal reduces repair efficiency.

The Britton *et al.*, 2014 study is the only other report directly indicating the formation of R-loops in human cells following DNA damage. The authors of that paper used a mutated version of RNase H1 fused to GFP to show a relocalisation of the enzyme to regions where DNA:RNA hybrids are formed following laser micro-irradiation. These studies however did not give any information about the nature of these structures, proximity to the break site or nucleotide-level detail about their distribution. DRIP-Seq has allowed us to examine the dynamics of DNA:RNA hybrid architecture around specific break-sites. Importantly, our DRIP-Seq data additionally demonstrate for the first time that these DNA:RNA hybrid structures are formed before DNA resection occurs, as otherwise the resected single stranded DNA would have been digested away during the process of library preparation, and thus not sequenced directly at the cut site.

In addition, we have now included examples of DRIP-Seq at different classes of DSB sites to highlight distinctive characteristics of genomic environments and provide new insights into the relationship between DNA:RNA hybrids, transcription and DSBs (Fig. S15). Our new data show that while DNA:RNA hybrids are forming around a large proportion of DSBs (55.6% show increased signal; 38.4% show above a 2-fold increase), not all cut sites show an induction of DNA:RNA hybrids suggesting their formation is regulated. Interestingly, while there is a clear requirement for transcription with RNA invasion occurring around the highest 2/3 of transcriptionally active cut sites but not the lowest 1/3 (Fig S15A), transcriptional activity does not automatically predispose a break site to DNA:RNA hybrid induction (Fig S15B), again showing that this process is regulated.

We have amended the text on page 9: *“Since the DRIP-Seq library preparation digests away any ssDNA, this shows that the DNA:RNA hybrid forms prior to resection and the resolution of the break. Importantly, the depletion of Drosha abrogates the enrichment of R-loops around break sites (Fig. 7C, S14BCD). It is unclear if the observed hybrid structures are the result of de novo synthesis following DNA damage or the increased interaction of pre-existing RNA molecules with their DNA template, but transcriptional activity of a locus prior to damage appears to be necessary for R-loop formation after damage (Fig. S15A). Curiously, observations at individual loci suggest transcriptional activity is not the sole determinant of damage-induced DNA:RNA hybridisation (Fig. S15B), indicating that the process is actively regulated following DNA damage.”*

We have also highlighted the novelty of our findings with text changes, including on page 10: *“We directly demonstrate for the first time in mammalian cells that DNA:RNA hybrids (R-loops) form around DSBs to facilitate repair, as was recently observed in *S.pombe*³⁸, and that this is Drosha-dependent either for their formation or retention (Fig. 7). We show that DNA:RNA hybrid formation is a very early event that precedes resection. Removal of DNA:RNA hybrids by RNase H1 overexpression results in impaired DNA damage repair (Fig. 7DE).”*

B) The reviewer asked us to suggest a feasible model for Drosha influence on R-loop

formation. We revised our manuscript including possible models for the role of Drosha and DNA:RNA hybrids in DNA repair (Fig S17). We also added text changes on page 11:

“Intriguingly, a recent publication characterised a resection-dependent end joining pathway in G1 phase of the cell cycle⁵⁰. It has been speculated that in the absence of a sister chromatid, the error-free repair in this pathway could only occur in the presence of an RNA template⁵¹, which would require hybridisation to the DNA around the break site (Fig. S17). The discovery of this novel repair pathway may explain our observation that DNA:RNA hybrid formation around DSBs occurs at sites prone to both NHEJ and HR or these structures could be a more general feature of DNA damage repair.” and *“It remains to be seen whether Drosha is directly involved in the Rad52-mediated RNA strand invasion, as reported in⁵⁴. This*

report suggested that Rad52 mediates RNA:DNA hybrid formation as a first step in an RNA-templated repair pathway.”

Major concerns

1. The reviewer felt that the experimental data regarding a lack of novel small RNA around DBSs did not include sufficient quality controls.

The purpose of our manuscript was not to persuade the reader that small RNA around break sites do not exist or that previously published work is incorrect, this as the reviewer points out, is a nearly impossible task. Our manuscript presents the use of a very different model system, in which we can examine multiple endogenous cut sites in different genomic settings. Previous reports have used single integrated DSB sites (Francia et al, 2012; Wei et al, 2012), not endogenous ones, and/or DSB sites within repetitive sequences (Iannelli et al, 2017). We show that Drosha has as strong an effect on the recruitment of repair machinery to AsiSI cut sites as it has in the context of IR-induced damage (Fig.1, 2 and Fig. S6), and thus this system allows for efficient and in-depth examination of changes in the RNA landscape. To address the reviewer’s comments we have now included extra data from our exhaustive search for small RNAs (Fig S11, S12, S13) detailed below:

Different sequencing approaches undertaken in the preparation of our manuscript:

Our first small RNA-Seq, the data for which we did not include in our original manuscript and have decided to leave out of the resubmission as well, we used U2OS cells stably transfected with the AsiSI endonuclease expressing plasmid with or without OHT induction. We selected a time of 6h after induction for sequencing, which was an earlier time point than other publications (Francia et al, 2012; Wei et al, 2012). This decision was based on our observation that Drosha acts in DNA damage very early (Figs 3, S5C). We failed to identify any small RNAs around the DSB sites using this approach (data not shown). There were a number of possible explanations for this: 1. the time point used; 2. different sequencing kit used; 3. the fact that the AsiSI enzyme is expressed also in control conditions could have resulted in a low level of “leakage” into the nucleus without induction. As a result, we decided to take the experimental approach described in the original manuscript. The experiments presented in the original manuscript were performed at multiple time points, including ones as long as those from previous publications, and these

also failed to identify any small RNAs (Fig. 6, S11A-C). The use of multiple time points meant that not only did we address the temporal aspect of the mechanism, but also effectively created multiple technical and biological replicates of these experiments. Following this, we hypothesised that the small RNAs could bear a 5' triphosphate, as seen during the ping-pong cycle for small RNA amplification in plants (Pak & Fire, 2007; Sijen et al, 2007), which would have prevented them from being readily detected in a standard sequencing approach. We therefore derived a method for sequencing 5' triphosphate small RNA in humans (Fig. S13). Analysis of these data also showed no small RNAs of this class generated from around break sites. These data are now included in the manuscript to show our exhaustive approaches to identify a class of small RNAs generated from around DNA break sites.

We have now added text to our manuscript to describe this on page 7: *“We also examined the size distribution of RNAs sequenced in our experiment (Fig. S11F), which confirmed that the vast majority of small RNAs corresponded to known annotated small RNAs. We additionally analysed RNAs both smaller (19-20nt) and larger (24-26nt) in size than the canonical 21-23nt, but this did not reveal any DSB-linked sequences over control conditions (Fig. S12). Finally, we excluded the possibility that the small RNAs could be generated with a 5'-triphosphate (as described in plants and lower eukaryotes^{33, 34}) (Fig. S13).”*

Sensitivity and Possible pitfalls

With respect to the comments about sensitivity, we would argue that the experiments we performed were at least as sensitive as those performed in previous publications. We have compared the sensitivity of our data set to that published in Francia et al. With the published data being derived from mouse cells and ours from human, the only quantifiable way of achieving this was to examine the raw reads for all conserved mature miRNAs. While the libraries had comparable numbers of reads (36.5M reads in REF14 data, 17M+67.7M in our CS data), our data clearly show that the distribution of reads per million (RPM) in our two sequencing experiments for conserved miRNAs has a less skewed distribution with fewer unrepresented miRNAs (Fig. S11D), compared to their single sequencing run (Francia et al, 2012). Indeed, direct comparison between 25 randomly selected conserved miRNAs shows no obvious bias between our data sets and the previously published results (Fig S11E).

This demonstrates that our datasets had higher depth and the use of biological replicates allowed us to perform more robust statistical testing of our data. With recently published single cell small RNA sequencing data showing that a cell only has around 3000 total miRNA molecules (Faridani et al, 2016), it should be possible to detect even a single novel small RNA molecule per cell. In terms of the sequencing platform, we utilised the NextSeq500 which outputs 400M reads across 12 samples. With small RNA sequencing, a single read covers the entirety of the original RNA, meaning that those 400M reads equal 400M directly sequenced RNA molecules. When coupled with our use of the AsiSI system – which we know cuts at 99 loci – we have skewed the odds in our favour but still did not detect anything above background. The size distribution of sequenced fragments shows the vast majority are in the 21-24nt range (Fig. S11F), which are almost exclusively mature miRNAs.

We have now added text to our manuscript to describe this on page 7: *“We compared the quality of our data to previously published high throughput experiments reporting the existence of DSB-derived small RNA¹⁴. As the previously published experiments had been performed in mouse cells, and ours in human cells, we were only able to directly compare conserved mature miRNAs. Our experiments show as good coverage and depth (Fig. S11DE) as that of the previous study, thus we were confident that small RNAs arising from DSBs should be robustly detected.”*

The reviewer states that they are not more or less convinced about the existence of small RNAs around DSBs.

While this was and still is not our primary objective, we hope that the additional data outlined above and included in the manuscript has addressed the reviewer’s concerns about the absence of small RNAs within our experimental system. Interestingly, we had not previously referenced a recently published paper, which also shows a lack of small RNA around DNA damage break sites in plants, and suggests that they are artefacts resulting from the repetitive nature of the genomic loci in which the DSB sites were situated (Miki et al, 2017). We have now made reference to this paper.

We have now stated on page 7: *“Indeed, a recent publication questioned the production of damage-induced small RNAs in plants suggesting their initial discovery may have resulted from analysis of particular reporter systems³⁵.”* and in the discussion, page 10: *“In addition, recent findings in plants show that small RNA could only be detected from a reporter locus but not endogenous loci³⁵.”*

2. AsiSI cleavage efficiency has not been determined and may change following Drosha depletion.

Assessment of this was originally omitted as previously it has been shown that the cleavage efficiency was high in the AsiSI system (~25%, Fig. S3B) (Aymard et al, 2014). This method relies on ligation of a biotinylated oligo duplex with a compatible AsiSI overhang to the cleaved DNA and therefore this approach will only detect unprocessed cleaved sites. In our system, we observe an efficiency of between 10%-25% (Fig. S9). Importantly, we have now also included data showing that the depletion of Drosha does not decrease the AsiSI cleavage efficiency; in fact, we find a slightly higher level of cleavage after Drosha depletion (Fig. S9), consistent with the protein’s role in the repair process (Fig. 4). This further supports the role of Drosha in repair and we would like to thank the reviewer for this comment. Consistent with this, our new quantitation of the number of γ -H2AX foci per cell confirms the average number of damage sites to be around 25-30, see Fig. S6B (based on the fact that 99 sites known to be cut in this system) (Aymard et al, 2014).

We have now added text to our manuscript to describe this, page 5 *“We also confirmed that depletion of Drosha does not alter the cleavage efficiency within the inducible system (Fig. S9).”*

3. “The TNRC6A-B knock-down suggest that Drosha effect is likely not mediated by the classical miRNA pathway, however, TNRC6C could potentially also mediate miRNA effects.”

The reviewer is correct, although TNRC6A-B depletion was sufficient to abolish miRNA-mediated repression (Fig. S1DE) suggesting that, as in other systems TNRC6C does not contribute significantly to repression, presumably due to its low expression levels

(Huntzinger et al, 2010). In the original manuscript we included data to show that depletion of TNRC6C together with TNRC6A and B does not affect the HR repair reporter (Fig. 4AB). We have now included additional data showing that TNRC6A, B and C depletion does not affect NHEJ repair reporter efficiency either (Fig. 4CD).

We have now added text to our manuscript to describe and clarify this point on page 5
“Similarly, a GFP reporter system, in which NHEJ-mediated repair of I-SceI cleavage leads to measurable expression of the reporter protein²⁹ (Fig. 4C), showed that Drosha and Dicer, but not TNRC6A-C, are also involved in the NHEJ pathway (Fig. 4D). This is consistent with our findings that the miRNA biogenesis enzymes function upstream of the divergence of the two pathways.”

4. Only a subset of AsiSI cut sites are cut within the genomic setting, how are these sites different and what does this mean for the experiments.

We thank the reviewer for raising this point and have now added text to our manuscript to discuss this as detailed below. In short, the reviewer is correct that sites actually cut by AsiSI are mainly found around promoters, partially due to the AsiSI recognition sequence (GCGATCGC). The methylation status and accessibility of these sites will both be contributing factors to their preferred cleavage. We agree that the proximity to promoters may result in increased small RNA background reads (Affymetrix & Cold Spring Harbor Laboratory, 2009; Kapranov et al, 2007; Taft et al, 2009). This is *precisely* why we felt it was critical that promoter areas with similar transcription activity serve as negative controls in our analysis (Fig. 6). Our data clearly show that Drosha depletion affects recruitment of repair machinery to AsiSI sites and if the repair was dependent on small RNAs, we should be able to detect them in this system around cut sites in the presence of Drosha, as discussed above. The baseline level of small RNA reads from the empty vector control transfection around these sites is very low (5.3 rpm over a 10kb window; for comparison, a 10kb area around an average miRNA gene will yield 718.8 rpm and our control uncut sites have 4.3 RPM) and thus not sufficient to prevent detection even if this was to occur at one cleavage site per cell.

In addition, the x-axes of the “random genomic site” graphs in Fig. 6 have been corrected from “AsiSI” to “0” which denotes non-cut sites with matched transcriptional activity. The

figure legend has been modified to say: *“These graphs are centred around a point (denoted as 0) at the same distance from the TSS as the matched AsiSI site.”*

We have added text on page 5: *“It should be noted that only a subset of possible AsiSI recognition sites in the genome were shown to be actually cut, most likely due to their accessibility and methylation status⁷.”*

Reviewer 2

We would like to thank the reviewer for these comments and in particular for suggesting a number of experiments to further our mechanistic understanding of the role of Drosha in DNA damage repair.

1. How does Drosha and Dicer function in establishing R-loops at DSB repair sites? Is R-loop formation required for ubiquitin signalling affected, for instance involving RNF8/RNF68?

As per our reply to reviewer #1, our study shows for the first time Drosha is involved in the maintenance of DNA:RNA hybrid structures at proximity of DSBs at a nucleotide resolution and, importantly, that:

- they form before the DNA is resected,
- their generation is Drosha dependent,
- they occur at both HR and NHEJ prone sites,
- they occur in the immediate proximity of the break site,
- their removal reduces repair efficiency.

Importantly, from our DRIP-Seq experiments and DNA end-resection assay we know that DNA:RNA hybrid formation occurs early and precedes the processing of DSB ends. In addition, we have now examined the recruitment of RNF168 and have found that it is also affected by Drosha depletion, further narrowing down the stage at which repair is affected (Fig. S6EF). Given the requirement of RNF168 for the recruitment of 53BP1, this explains the marked reduction of 53BP1 foci formation upon Drosha knockdown (Fig. 1,2,3,S5C). This also fits the model where Drosha is involved in chromatin remodelling after DNA damage. One would expect the downstream Histone H2A ubiquitylation to be implicated (Bohgaki et al, 2013; Fradet-Turcotte et al, 2013). Whether upstream Histone H1 ubiquitylation

(Thorslund et al, 2015) or other histone modifications are perturbed will be interesting future directions.

We have concentrated on Drosha's role as this is the main focus of the manuscript and we have expression constructs to conduct these experiments relatively quickly. To assay whether Drosha's RNase activity was required for DSB repair, we generated a Drosha mutant with a two amino acid substitution at its RNase III domains. This mutant was reported to abolish Drosha's ability to process miRNA precursors (Han et al, 2004). We attempted to overexpress the mutant and WT Drosha constructs to assess the cells' ability to resect DNA ends in response to DSB. However, the overexpression of mutant Drosha appears to be highly toxic in our U2OS-AsiSI system and thus no conclusions could be drawn. We thank the reviewer for the suggestion and this should be the future direction of the investigation.

However, we now include data to shown that rescue of wild-type Drosha restores 53BP1 foci and resection (Fig. S7G-I and S10A).

We have added text to explain this new data on page 5-6: *"The recruitment of the E3 ubiquitin ligase RNF168 was also examined and showed reduced foci formation upon depletion of Drosha (Fig. S6EF), thus strengthening the conclusion that Drosha acts at the chromatin remodeling phase prior to 53BP1 recruitment."*

2. Is the removal of R-loops (by overexpression of RNase H1) epistatic with Drosha depletion?

This is a very interesting question and we have conducted this experiment using the DNA resection assay for two HR-repaired sites. This shows that overexpression of RNase H1 together with Drosha knockdown does not further decrease R-loop formation beyond knockdown of Drosha alone (Fig S16DE).

We have added text to explain this new data on page 9: *"Combining this with depletion of Drosha did not have an additive effect on ssDNA resection, strongly suggesting that Drosha acts to promote the formation of R-loops (Fig. S16DE)."*

3. The reviewer asks if the effects of Drosha and Dicer are restricted to HR or also affect the NHEJ pathway.

We have now added additional data showing that the NHEJ pathway is also affected, using GFP reporter assays (Seluanov et al, 2010) (Fig. 4CD). This is in agreement with the effect we observe on 53BP1 recruitment (Fig. 1) and with the data that shows sites undergoing predominantly NHEJ repair also accumulate R-loops (Fig. 7,S14CD).

We have added text to describe these new data on page 5: *“Similarly, a GFP reporter system, in which NHEJ-mediated repair of I-SceI cleavage leads to measurable expression of the reporter protein²⁹ (Fig. 4C), showed that Drosha and Dicer, but not TNRC6A-C, are also involved in the NHEJ pathway (Fig. 4D). This is consistent with our findings that the miRNA biogenesis enzymes function upstream of the divergence of the two pathways.”*

We have also amended the abstract to clearly state this point: *“Depletion of Drosha significantly reduces DNA repair by both homologous recombination (HR) and non-homologous end joining (NHEJ).”*

4. “I find the term ‘DDR positive cells’ misleading. It would be better to indicate absolute numbers of foci per cell in Figures 1, 2 and 3.”

We have now reanalysed all the DNA damage foci data within our manuscript using the ImageJ FindFoci plug-in developed by the Hoffman lab at University of Sussex (Herbert et al, 2014). This has provided us with an unbiased quantitation of numbers of foci per cell and has replaced previous data in Figs. 1B, 2B, 3B, S5C, S6BDF, and S7BE within our revised manuscript. We would like to thank the reviewer for this suggestion, and believe this strengthens our results.

5. The reviewer asks “Figure 3: the kinetics of 53BP1 recruitment are different between foci and laser stripe. It seems that in the kinetic analysis....”

These are very different DNA damage inducers, as I am sure the reviewer knows. To give a more comprehensive view of the temporal response of repair signalling to Drosha depletion, we have now re-quantified and collated our data of 53BP1 recruitment to foci over multiple

time points (Fig S5C). This shows that 53BP1 recruitment is impaired at all time points investigated following Drosha knockdown.

We have added text to describe this on page 4: *“Consolidation of our data from different time points confirmed that impairment of 53BP1 recruitment is an early event and continues for long periods of time following exposure to DNA damage (Fig. S5C).”*

6. We are sorry that the figure legend Fig 5C resulted in a misunderstanding. The data are not normalised to the miRNA site, but rather to an independent ChIP of histone H3 for both the miRNA locus and the DSB site.

We have adjusted the figure legend of Fig. 5C to make this clearer *“qPCR of Drosha and control IgG ChIP at an HR locus and a canonical Drosha binding site at the miR-122 genomic locus 1 hour after damage induction. The ChIP efficiency was calculated against a histone H3 ChIP performed in parallel, error bars=SEM, Student’s 2-sample T-test, **p≤0.01, N=3.”*

7. “What is the consequence of Drosha or Dicer inactivation for cells? Does it lead to more mutations, translocations, or sensitivity to DNA damaging agents?”

We have now conducted additional experiments addressing this question and include data showing that depletion of Drosha and Dicer both increase entry into late apoptosis following DNA damaging agent bleomycin (Fig. S5D).

This data is now included and text on page 4 has been modified to read *“Accordingly, Drosha and Dicer depletion increases the entry into late apoptosis following radiomimetic bleomycin-induced DNA damage (Fig. S5D).”*

8. “- Page 3/line 76: ‘This confirmed that...’ this is overstated. Should say that it is likely not linked to the canonical function.”

We have corrected this.

9. “-page 4/line 116: paragraph title is weird: is it required for HR? or proficient HR? please change.”

We have changed this to “effective HR and NHEJ” (due to new data having been added).

Reviewer 3

We would like to thank the reviewer for their comments and for all the suggestions which have added significantly to our manuscript.

1. Different siRNAs should be used to deplete Drosha and Dicer, and rescue experiments should be used for Drosha.

We agree and apologise that this data we not included in the first submission. We have now included additional data with alternative Drosha and Dicer siRNAs (Fig. S7A-F) and have examined both 53BP1 foci formation and DNA strand resection following siRNA rescue experiments with the over-expression of an siRNA-resistant Drosha (Fig. S7G-I and S10A).

We have changed the text to reflect these new data on page 5: *“We confirmed that these effects were unchanged using different siRNAs against Drosha and Dicer (Fig.S7A-F). Rescue experiments using an siRNA-resistant over-expression plasmid showed that the effects on 53BP1 recruitment were specific to Drosha knockdown (Fig.S7GHI).”*

and page 6 *“This could be rescued by the over-expression of an siRNA-resistant Drosha (Fig. S10A), corroborating results of the rescue experiment which showed a restoration of 53BP1 foci (Fig. S7GHI).”*

2. Double KD of Drosha and Dicer should be conducted to determine if they are epistatic.

We have conducted these experiments using the NHEJ GFP reporter system (Fig. 4CD) as well as the HR site resection assay (Fig. S10BC). In each of these approaches we observe that while depletion of both proteins inhibits repair efficacy, the double depletion results in a similar degree of repair impairment as the strongest single knockdown. Thus, it seems that the proteins do not act redundantly and participate in the same pathway. It remains to be seen if they act sequentially and in what order.

The text has been changes to reflect this additional data on page 6: *“We also used this assay to confirm that knockdown of Dicer had a similar effect on resection as Drosha depletion (Fig.S10BC). The two proteins seem not to have redundant functions in this pathway, as a*

double knockdown of Drosha and Dicer together has the same effect on resection as depletion of only one of the proteins (Fig. S10BC)."

3. Quantification of foci and evaluating if we observe a mixture of weak and bright foci following Drosha/Dicer depletion, could this indicate a change in recruitment kinetics.

Unfortunately, we do not have access to software that can resolve focus brightness. However, we have re-quantified all the data Fig. 1B, 2B, 3B, S5C, S6BDF, and S7BE. Visual examination of the effect of Drosha depleted on the brightness of foci did not indicate any consistent trend. Importantly, we have re-quantified 53BP1 foci at different time points following DNA damage and observe a reduced signal following Drosha depletion at each time point studied (Fig S5C).

4. The reviewer asks us to use a different method of foci quantification.

As stated above, we have re-quantified all of the DNA damage foci in Figs 1B, 2B, 3B, S5C, S6BDF, and S7BE using the ImageJ FindFoci plug-in. We thank the reviewer for this suggestion and we believe this strengthens our observations.

5. Fig 2. Title is an overstatement. Many thanks we have changed the figure title to read *"Drosha is involved in DDR downstream of MDC1."*

6. "Lines 137/138 of main text. Please check Legube paper (ref 7)."

Many thanks we have changed text to read *"They were able to map the endogenous cut sites and determine which individual sites were predominately utilizing either HR or NHEJ thanks to preferential association of Rad51 with HR sites⁷."*

7. Fig S2. Do Drosha KD cells really have more intense ATMph foci?

We do not believe this is the case. Since the threshold for our quantitation of phosphoATM foci is set in the control conditions, a consistently higher intensity in the Drosha KD would have likely resulted in a higher number of foci passing the threshold, which is not the case. Also, visual assessment confirms they are not more intense and our phosphoATM Western blots also show no significant differences in ATM phosphorylation (Fig. 1C, S8C, S10B, S16E).

In order to highlight this we have replaced Figure S2 with a more representative image. We thank the reviewer for pointing this out.

8. “As Dicer and Drosha are required to generate miRNAs, why does their KD have no effect on Let-7-dependent miRNA mediated repression? Please clarify.”

MiRNAs mainly have relatively long half-lives and over the timespan of these experiments, depletion of Drosha and Dicer do not significantly affect miRNA levels (Fig S1C). This allowed us to separate canonical miRNA activity from the effect we observe on DNA damage foci following Drosha and Dicer depletion.

We have changed the text to make this clearer on page 3: *“As expected, given the longevity of some small RNAs²², the depletion of the biogenesis enzymes for 48 h did not result in changes in mature miRNA levels explaining why Drosha/Dicer depletion does not affect miRNA-mediated repression (Fig. S1E).”*

9. Line 157 replace “profound” with “significant”.

Done, many thanks.

10. “The authors should examine other size ranges within their data sets, e.g. 19-20 nt and 24-26 nt. We would like to thank the reviewer for this excellent idea. We have examined the size range indicated and have included additional figures (Fig S11F, S12) and have changed the text accordingly. This analysis shows that after the exclusion of miRNAs and other annotated small RNA, RNAs in the 21-23 nt range constitute ~20% of small RNA reads sequenced (Fig. S11F, right panel). Nevertheless, small RNAs outside that range (19-20 and 24-26 nt) also show no trace of DNA damage-induced small RNA (Fig. S12AB). For the 24-26 nt range, 2 AsiI sites (1 cut and 1 uncut) pass significance threshold (Fig. S12B). However, in the case of the site known to be cut (site89, Fig S12C), this change is a decrease; and in the case of the site known to not be cut (non412, Fig S12D), the significant change is directly adjacent to an annotated Y RNA and thus almost certainly a misannotation.

We have changed the text on page 7: *“We also examined the size distribution of RNAs sequenced in our experiment (Fig. S11F), which confirmed that the vast majority of small RNAs corresponded to known annotated small RNAs. We additionally analysed RNAs both*

smaller (19-20nt) and larger (24-26nt) in size than the canonical 21-23nt, but this did not reveal any DSB-linked sequences over control conditions (Fig. S12)."

11. Investigate mCherry-RNase H1 recruitment upon depletion of Drosha/Dicer.

Many thanks for the excellent suggestion. We have conducted the experiments and the removal of Drosha and Dicer reduces the recruitment of mCherry-RNaseH1^{mut} to sites of irradiation. This new data has now been added in Fig.S16A.

The text has been adjusted accordingly on page 5: *"Furthermore, depletion of both Dicer and Drosha resulted in a significant delay in the relocalisation of RNase H1 to the site of laser irradiation (Fig. S16A), indicating that both proteins have a role in the formation of DNA:RNA hybrids after DNA damage induction."*

12. "Fig 7C. Very nice DNA:RNA hybrid DRIP-seq data in human cells that is supported by recently published observations in yeast (ref 35). Regarding the issue of de novo transcription, have the authors considered using RNA pol II inhibitors? I appreciate the expense of these experiment, so I do not consider this to be essential for publication".

While this is an excellent idea, the DRIP-Seq experiments would be very difficult and costly to perform, especially within our current time constraints. We attempted to perform mutant RNase H1-mCherry laser track experiments after treatment with α -amanitin. Unfortunately, the treatment made the cells so sensitive to the laser, that creating the stripe resulted in their physical destruction. Of the very few cells that did survive, no recruitment could be observed however the number of cells obtainable was far too low to include within a manuscript.

Additional analysis of our existing DRIP-Seq data shows that R-loops do not accumulate within less transcriptionally active regions of the genome (Fig S15A).

We have added text on page 9: *"It is unclear if the observed hybrid structures are the result of de novo synthesis following DNA damage or the increased interaction of pre-existing RNA molecules with their DNA template, but transcriptional activity of a locus prior to damage appears to be necessary for R-loop formation after damage (Fig. S15A). Curiously, observations at individual loci suggest transcriptional activity is not the sole determinant of*

damage-induced DNA:RNA hybridisation (Fig. S15B), indicating that the process is actively regulated following DNA damage.”

13. Fig 7D. Also a very nice preliminary experiment consistent with a positive role for R-loops in repair by HR. Enforced removal of R-loops by expression of RNase H1 abrogates resection and again consistent with the recent yeast data. Perhaps defective repair in the Jasin assay following expression of RNase H1 could be used to independently support the data with AsiSI sites.

We have now conducted the suggested experiment using the HR and NHEJ GFP repair reporter constructs combined with over-expression of RNaseH1. This resulted in a statistically significant reduction in repair efficiency in each case (Fig 7E, S16F). We would like to thank the reviewer for this suggestion.

We have changed the text to support this new data on page 9: “Over-expression of RNase H1 is currently the only way to investigate the consequences of R-loop formation in vivo⁴² and we confirmed that this resulted in a decrease in both HR and NHEJ repair efficiency using the GFP reporter systems (Fig. 7E, S16F).”

14. “Label for Y axis missing from Fig S7A & B”

Added – these are now S8A and B; very sorry and thanks for pointing this out.

Bibliography

Affymetrix ETP, Cold Spring Harbor Laboratory ETP (2009) Post-transcriptional processing generates a diversity of 5'-modified long and short RNAs. *Nature* **457**: 1028-1032

Aymard F, Bugler B, Schmidt CK, Guillou E, Caron P, Briois S, Iacovoni JS, Daburon V, Miller KM, Jackson SP, Legube G (2014) Transcriptionally active chromatin recruits homologous recombination at DNA double-strand breaks. *Nature structural & molecular biology* **21**: 366-374

Bohgaki M, Bohgaki T, El Ghamrasni S, Srikumar T, Maire G, Panier S, Fradet-Turcotte A, Stewart GS, Raught B, Hakem A, Hakem R (2013) RNF168 ubiquitylates 53BP1 and controls its response to DNA double-strand breaks. *Proceedings of the National Academy of Sciences of the United States of America* **110**: 20982-20987

Britton S, Deroncourt E, Delteil C, Froment C, Schiltz O, Salles B, Frit P, Calsou P (2014) DNA damage triggers SAF-A and RNA biogenesis factors exclusion from chromatin coupled to R-loops removal. *Nucleic acids research* **42**: 9047-9062

Faridani OR, Abdullayev I, Hagemann-Jensen M, Schell JP, Lanner F, Sandberg R (2016) Single-cell sequencing of the small-RNA transcriptome. *Nature biotechnology* **34**: 1264-1266

Fradet-Turcotte A, Canny MD, Escibano-Diaz C, Orthwein A, Leung CC, Huang H, Landry MC, Kitevski-LeBlanc J, Noordermeer SM, Sicheri F, Durocher D (2013) 53BP1 is a reader of the DNA-damage-induced H2A Lys 15 ubiquitin mark. *Nature* **499**: 50-54

Francia S, Michelini F, Saxena A, Tang D, de Hoon M, Anelli V, Mione M, Carninci P, d'Adda di Fagagna F (2012) Site-specific DICER and DROSHA RNA products control the DNA-damage response. *Nature* **488**: 231-235

Han J, Lee Y, Yeom KH, Kim YK, Jin H, Kim VN (2004) The Drosha-DGCR8 complex in primary microRNA processing. *Genes & development* **18**: 3016-3027

Herbert AD, Carr AM, Hoffmann E (2014) FindFoci: a focus detection algorithm with automated parameter training that closely matches human assignments, reduces human inconsistencies and increases speed of analysis. *PLoS one* **9**: e114749

Huntzinger E, Braun JE, Heimstadt S, Zekri L, Izaurralde E (2010) Two PABPC1-binding sites in GW182 proteins promote miRNA-mediated gene silencing. *The EMBO journal* **29**: 4146-4160

Iannelli F, Galbiati A, Capozzo I, Nguyen Q, Magnuson B, Michelini F, D'Alessandro G, Cabrini M, Roncador M, Francia S, Crosetto N, Ljungman M, Carninci P, d'Adda di Fagagna F (2017) A damaged genome's transcriptional landscape through multilayered expression profiling around in situ-mapped DNA double-strand breaks. *Nature communications* **8**: 15656

Kapranov P, Cheng J, Dike S, Nix DA, Duttgupta R, Willingham AT, Stadler PF, Hertel J, Hackermuller J, Hofacker IL, Bell I, Cheung E, Drenkow J, Dumais E, Patel S, Helt G, Ganesh M, Ghosh S, Piccolboni A, Sementchenko V, Tammana H, Gingeras TR (2007) RNA maps reveal new RNA classes and a possible function for pervasive transcription. *Science* **316**: 1484-1488

Miki D, Zhu P, Zhang W, Mao Y, Feng Z, Huang H, Zhang H, Li Y, Liu R, Zhang H, Qi Y, Zhu JK (2017) Efficient Generation of diRNAs Requires Components in the Posttranscriptional Gene Silencing Pathway. *Scientific reports* **7**: 301

Pak J, Fire A (2007) Distinct populations of primary and secondary effectors during RNAi in *C. elegans*. *Science* **315**: 241-244

Seluanov A, Mao Z, Gorbunova V (2010) Analysis of DNA double-strand break (DSB) repair in mammalian cells. *Journal of visualized experiments : JoVE*

Sijen T, Steiner FA, Thijssen KL, Plasterk RH (2007) Secondary siRNAs result from unprimed RNA synthesis and form a distinct class. *Science* **315**: 244-247

Taft RJ, Glazov EA, Cloonan N, Simons C, Stephen S, Faulkner GJ, Lassmann T, Forrest AR, Grimmond SM, Schroder K, Irvine K, Arakawa T, Nakamura M, Kubosaki A, Hayashida K, Kawazu C, Murata M, Nishiyori H, Fukuda S, Kawai J, Daub CO, Hume DA, Suzuki H, Orlando V, Carninci P, Hayashizaki Y, Mattick JS (2009) Tiny RNAs associated with transcription start sites in animals. *Nature genetics* **41**: 572-578

Thorslund T, Ripplinger A, Hoffmann S, Wild T, Uckelmann M, Villumsen B, Narita T, Sixma TK, Choudhary C, Bekker-Jensen S, Mailand N (2015) Histone H1 couples initiation and amplification of ubiquitin signalling after DNA damage. *Nature* **527**: 389-393

Wei W, Ba Z, Gao M, Wu Y, Ma Y, Amiard S, White CI, Rendtlew Danielsen JM, Yang YG, Qi Y (2012) A role for small RNAs in DNA double-strand break repair. *Cell* **149**: 101-112

Reviewers' Comments:

Reviewer #1:

Remarks to the Author:

The authors have answered all my questions and concerns. The manuscript and the presented data are significantly improved and I recommend it for publication.

I have only one question/comment for the authors:

Figure S15A – "...but transcriptional activity of a locus prior to damage appears to be necessary for R-loop formation after damage (Fig. S15A)" – since the AsiSI cleavage is quite selective, it is not clear whether transcriptionally less active sites are also less efficiently cleaved by the enzyme. The conclusion of this figure is only valid if the authors show that this "low transcriptional activity" subset of cleavage sites are cleaved with similar efficiency to the medium or highly transcribed sites. The authors might have this data, since they determined the cleavage efficiency of the potential AsiSI sites. If they don't have the cleavage efficiency data for all cleavage sites, I would consider omitting this figure and the conclusion that only transcriptionally active sites form DNA-RNA hybrids. This is a potentially important observation to determine if DNA-RNA hybrids are formed by de novo transcription or by RNAs transcribed before the DNA damage occurred. Also, looking at the individual loci in the DRIP-seq data (before DNA damage), raises some questions about the quality of this dataset (S15B). It doesn't show the typical DNA-RNA hybrid peaks around termination sites and promoters and doesn't seem to show correlation with transcriptional activity of the DNA region (average DRIP FPM seems to be very similar between transcribed and non-transcribed regions), which contradicts previously published results. It seems to me that the majority of the signal is background signal and it might be misleading to analyse this signal and draw strong conclusions from it. If I did not see these particular examples, I would not have picked it up, but I would recommend the authors reconsider some of the conclusions that they based on this data.

Reviewer #2:

Remarks to the Author:

In their revised manuscript, Lu and co-workers have provided additional experimental data to address concerns. Overall, the authors have added many additional control experiments, and have re-analyzed important experiments. Overall, the new data have strengthened the conclusions, although I still feel that mechanistically I don't understand what is going on.

- RNF168 is also affected, further narrowing down where the defects in DNA repair occur. However, this shifts the pathway 1 step up. A key experiment would be to investigate Histone ubiquitylation at break sites.

-the fact that DNA:RNA hybrids already form prior to DNA-end resection is a bit of an overstatement. This is, as the authors indicate, due to the theoretical inability of DRIP-seq to amplify regions where DNA is resected. DNA:RNA hybrid formation and DNA resection were not measured directly.

-the observation that R-loops are detected at many but not all break sites, and the observation that active transcription is observed at break sites with R-loops, makes the authors conclude that the process of DNA:RNA hybrids is regulated. Although that might be true, the data do not provide proof (might also be due to sequence-specific abilities (eg GC content, secondary structures)). Such statements should be down tuned.

-On page 11, the new text reads: "Intriguingly, a recent publication characterised a resection-dependent end joining pathway in G1 phase of the cell cycle⁵⁰. It has been speculated that in the absence of a sister chromatid, the error-free repair in this pathway could only occur in the

presence of an RNA template⁵¹, which would require hybridisation to the DNA around the break site (Fig. S17). The discovery of this novel repair pathway may explain our observation that DNA:RNA hybrid formation around DSBs occurs at sites prone to both NHEJ and HR or these structures could be a more general feature of DNA damage repair.”

According to the paper by the Lobrich lab, end-resection in G1 does not lead to HR, but to error-prone end-joining. This is not in line with the observation that DNA:RNA hybrids forms at sites prone to undergo HR. in line with their model, rad52 foci should be increased in drosha-depleted cells? To my opinion, the proposed model is not adequate.

Reviewer #3:

Remarks to the Author:

The authors have satisfactorily address all my major concerns in a highly considered and comprehensive manner. I support publication in Nature Communications.

Reviewer 1

This reviewer recommends our manuscript for publication, but additionally has one question/comment for the authors.

The reviewer states we need to provide evidence of that cut sites with low transcriptional activity have similar AsiSI cleavage efficiency to medium and high transcriptional activity sites (Fig.15A). We thank the reviewer for this comment and completely agree. Fig.S9 originally had the cleavage efficiencies for three sites and they were chosen as they represent a diverse range of transcriptional activities which we did not point out. In addition to this, we have now also included an additional site which has low transcriptional activity (Fig.S9). We observe a similar degree of cleavage efficiencies between these sites. However, we do not have cleavage efficiency data for all sites and therefore we have also downplayed our conclusions.

We have changed Fig.S9 to show transcriptional status of these sites, and have made adjusted text accordingly (page 9): “It is unclear if the observed hybrid structures are the result of *de novo* synthesis following DNA damage or the increased interaction of pre-existing RNA molecules with their DNA template, but transcriptional activity of a locus prior to damage appears to predispose a site to R-loop formation after damage (Fig. S15A), while it does not seem to affect their cleavage efficiency (Fig. S9).”.

The reviewer also felt that the quality of the DRIP-Seq data is low and suggests that we alter some of our conclusions. We understand the reviewer’s concerns and we have included additional data to show that our data shows the expected features of R-loop architecture. We have now included a metagene analysis that shows the expected relationship between R-loops and transcriptional start and termination sites (Fig.S14C). These results are similar to published data (Stork et al, eLife, 2016). We have adjusted the text accordingly (page8): “We confirmed that DNA:RNA hybrids accumulate in the promoter and termination regions of genes in a transcriptional activity-dependent manner, as seen previously (Fig.S14C)⁴²”.

Reviewer 2

This reviewer feels that the additional experiments and re-analysis has strengthened the manuscript, but asked for 1 additional experiment and alterations to the text, which we have now conducted.

1. The reviewer asks if histone ubiquitylation is affected at break sites. We find that following Drosha depletion there is a significant reduction of ubiquitin conjugation to chromatin (Fig.S6GH). These results show that proficient chromatin ubiquitination is dependent upon Drosha around DSBs. Precise identity of the histones differentially modified and the type of ubiquitination linkage involved is outside the scope of the manuscript. We have adjusted the text (page 5): “The recruitment of the E3 ubiquitin ligase RNF168 and the ubiquitination of chromatin was also reduced upon depletion of Drosha (Fig. S6E-H), thus strengthening the conclusion that Drosha acts at the chromatin remodelling phase prior to 53BP1 recruitment.)”.

2. The reviewer states that we cannot conclude that DNA:RNA hybrids are formed before resection as it is only a “theoretical inability” that the sequencing would not be able to pick-up resected DNA. We disagree with the reviewer on this point. This is actually a technical fact. Following IP of the DNA:RNA hybrid-containing fragment, the RNA component is removed and the fragment is end-repaired for downstream adaptor ligation. This step results in single-stranded 3’ overhangs (which would be the product of resection following DNA damage) being removed. Thus, any sequenced fragments would originate further from the break if resection had already occurred, and our data does not show this. Importantly, we have used a paired-end sequencing approach which allows us to accurately determine the defined ends of our read fragments. This is a point that may have been overlooked by the reviewers and we have now made reference to it in our manuscript.

The clear distribution of the DNA:RNA hybrid structures demonstrates that these sequenced fragments originated from unresected DNA, showing that RNA invasion occurs before resection. Additionally, the exonucleolytic cleavage of 3’ overhangs before adapter ligation in the library preparation protocol is very clearly demonstrated by the complete absence of sequencing reads that cover the 3’ overhang generated by AsiSI in our sequencing data sets (Fig S14D).

Other data included in this manuscript supports these observations, namely the resection assays in figures 5B, 7D and S10A, which show that both Drosha (which we show is required for DNA:RNA hybrid formation) and RNase H1 over-expression (which removes DNA:RNA hybrids) reduced resection efficiency, suggesting a requirement for RNA invasion prior to resection.

We have adjusted text to better relay this information (page 9): “Since the DRIP-Seq library preparation includes a step in which 3’-overhang ssDNA fragments are exonucleolytically digested, this shows that the DNA:RNA hybrid forms prior to resection and the resolution of the break. Importantly, the depletion of Drosha abrogates the enrichment of R-loops around break sites (Fig.7C, S14BEF).”.

3. The reviewer states that we don’t have proof for the DNA:RNA hybrids being regulated around break sites. We understand the reviewer’s point and agree. We have changed the text accordingly (page 9): “Curiously, observations at individual loci suggest transcriptional activity is not the sole determinant of damage-induced DNA:RNA hybridisation (Fig. S15B), indicating that other factors influence their formation.”.

4. The reviewer dislikes the model proposed. We have removed the model and adjusted text accordingly.

Reviewer 3

This reviewer is now satisfied and supports publication.

Reviewers' Comments:

Reviewer #1:

Remarks to the Author:

The authors have answered all my questions and concerns. The manuscript and the presented data are significantly improved and I recommend it for publication.

Reviewer #2:

Remarks to the Author:

I appreciate the textual changes in the manuscript, especially those that make it easier to grasp the experimental procedures on the DNA sequencing.

Concerning mechanistic insight, the authors have looked at ubiquitination, as this may provide insight into a link between Drosha and RNF8/RNF168 activity at break sites. Immunofluorescence analysis using the FK2 antibody is used, which detects ubiquitin moieties connected to lysines, but not unconjugated ubiquitin. The results indicate that Drosha depletion results in a decrease in FK2 foci.

In the rebuttal, the authors state that: "Precise identity of the histones differentially modified and the type of ubiquitination linkage involved is outside the scope of the manuscript."

I agree that an analysis of specifically how and which histones are differentially modified (although very insightful), may be beyond the scope of this study. However, this statement starts with the assumption that histones are differentially modified. Yet, the FK2 staining is known to also detect ubiquitin modification of non-histones, even non-chromatin factors. So, the rebuttal text is an overstatement of the data, and the newly added text in the results section on 'ubiquitination of chromatin' also is not very adequate. I would rephrase this to something like: 'local ubiquitination at DNA damage foci'.

Other than that, I find the quality of the entire manuscript sufficient to support publication.

Reviewer 2

This reviewer now supports publication but required one additional text change now embedded in the final version.

The reviewer required that we change the words “ubiquitination of chromatin” to “local ubiquitination at DNA damage foci”

We have now changed text as follows:

“The recruitment of the E3 ubiquitin ligase RNF168 and local ubiquitination at DNA damage foci was also reduced upon depletion of Drosha (Fig. S6E-H), thus strengthening the conclusion that Drosha acts at the chromatin remodeling phase prior to 53BP1 recruitment.”